# From Clutter to Clarity: Visual Recognition through Foveated Object-Centric Learning (FocL)

**Amitangshu Mukherjee** *mukher44@purdue.edu*
*Elmore Family School of Electrical and Computer Engineering*
*Purdue University*

**Deepak Ravikumar** *dravikum@purdue.edu*
*Elmore Family School of Electrical and Computer Engineering*
*Purdue University*

**Kaushik Roy** *kaushik@purdue.edu*
*Elmore Family School of Electrical and Computer Engineering*
*Purdue University*

**Reviewed on OpenReview:** *https://openreview.net/forum?id=kVS7sMlv7P*

## Abstract

Human active vision integrates spatial attention (dorsal) and object recognition (ventral) as distinct information processing pathways. Rapid eye movements focus perception on task-relevant regions while filtering out background clutter. Mimicking this ventral specialization, we introduce FocL (Foveated Object-Centric Learning), a training strategy that biases image classification models toward label-consistent object regions by replacing full images with foveated crops. Standard training often relies on spurious correlation between label and background, increasing memorization of hard examples in the tail of the difficulty distribution. FocL simulates saccades by jittering fixation points and extracting foveated glimpses from annotated bounding boxes. This object-first restructuring reduces non-foreground contamination and lowers mean training loss. FocL reduces memorization, lowering mean cumulative sample loss by approximately 65% and making nearly all high-memorization samples (top 1%) easier to learn. It also increases the mean $\ell_2$ adversarial perturbation distance required to flip predictions by approximately 62%. On ImageNet-V1, FocL achieves up to 11% higher accuracy on oracle crops. When paired with the Segment Anything Model (SAM) as a dorsal proposal generator, FocL provides up to 7% gain on ImageNet-V1 and up to 8% under natural distribution shift (ImageNet-V2). Extending this setup to COCO, FocL improves cross-domain mAP by 3–4 points without any target-domain training. Finally, given object localization (bounding boxes), FocL reaches higher accuracy using roughly 56% fewer training images, offering a simple path to more robust and efficient visual recognition. [1]

## 1 Introduction

Deep neural networks often achieve high performance by relying on spurious correlations between labels and irrelevant background features (Bayat et al., 2025; Geirhos et al., 2020), rather than learning robust object-centric representations. This hinders generalization on hard examples in the tail of the sample-level difficulty distribution, even when class frequencies are balanced (Arpit et al., 2017; Usynin et al., 2024).

---

[1]The code is available at GitHub.

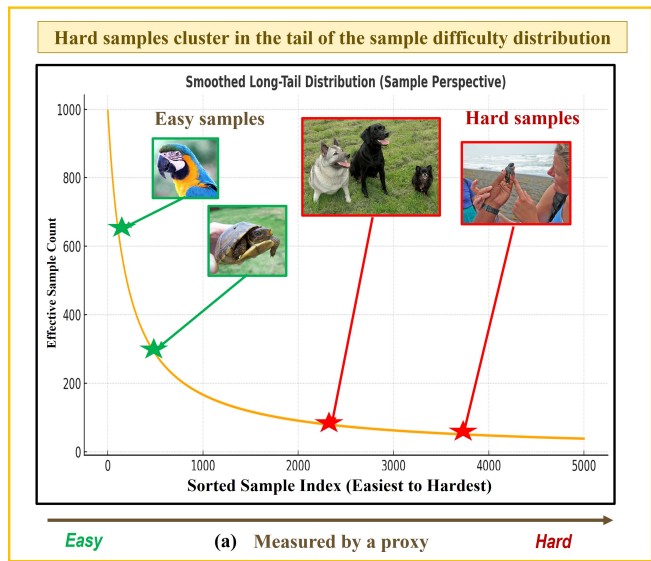
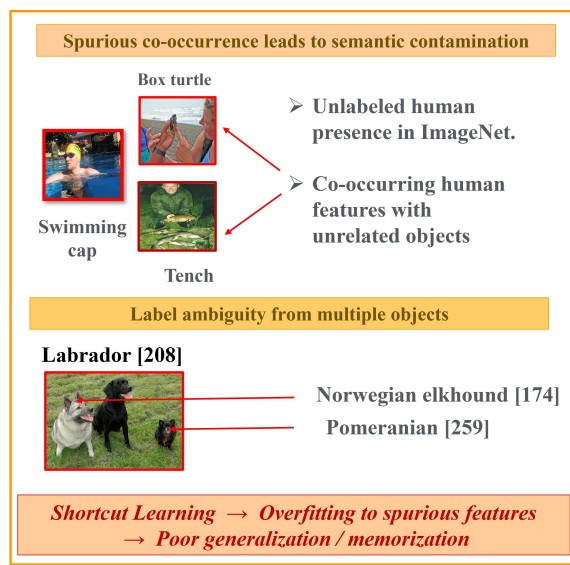

Figure 1: Figure illustrates key challenges that drive memorization and hinder generalization in visual recognition. (Left) A conceptual long-tail curve of sample-level difficulty, with harder examples concentrated in the tail and difficulty measured via proxies such as sample loss. (Right) Two major sources of sample-level hardness: (a) Spurious correlations from unlabeled co-occurring entities (e.g., humans) cause models to overfit to background context; (b) Label ambiguity from multi-object images (e.g., a "Labrador" sample also containing other dog breeds) introduces confusion. These effects weaken object-label consistency and promote reliance on shortcuts.

An example of sample-level difficulty is the sample's training loss (or its gradient norm), which quantifies how challenging it is to learn. The left sub-panel of Figure 1 illustrates this: harder examples concentrate in the tail under difficulty measures (e.g., loss or curvature) (Garg et al., 2024; Ravikumar et al., 2024; 2025). These instances often lead to memorization, where models overfit to background context, dataset artifacts, or unrelated co-occurring objects instead of focusing on the labeled foreground object (Brown et al., 2021; Feldman & Zhang, 2020). The right sub-panel of Figure 1 illustrates common failure sources. These failures include unlabeled distractors like humans, and label ambiguity from multiple objects in a single annotated image, for example, a "Labrador" image that also contains other dog breeds.

Existing methods mitigate learning challenges in long-tailed distributions through class re-balancing or data augmentation strategies (Kang et al., 2020; Ren et al., 2020; Tan et al., 2021; Yun et al., 2019; Zhang et al., 2018; 2023). However, these approaches primarily target class-level imbalance rather than sample-level difficulty. In contrast, we focus on individual examples that challenge a model even when class frequencies are balanced. This leads to a natural question: can generalization be improved by presenting object-centric, foveated views analogous to how humans focus on informative regions while filtering out irrelevant and spurious features?

To explore this, we draw on insights from biological vision, where focusing on task-relevant regions offers a natural analogy for object-centric learning. As illustrated in Figure 2(a), human perception operates through an active vision system that combines goal-directed sampling with object-centric encoding. The initial visual input is captured via peripheral vision, which provides coarse information across the scene. Based on this, saccadic eye movements shift the fovea, the high-acuity center of the retina, toward salient targets. According to the two-stream hypothesis (Clark, 2013; Goodale & Milner, 2004; Milner & Goodale, 1992; Mishkin et al., 1983; Sakuraba et al., 2012; Ungerleider & Haxby, 1994), the dorsal stream computes where to look by identifying spatially informative regions. In parallel, the ventral stream processes the high-resolution foveated input (Eckstein, 2011; Shao et al., 2024) to determine what is being observed, extracting semantic features such as shape and identity. This foveated mechanism allows humans to extract consistent, object-centered representations across varied contexts, forming the basis for robust generalization.

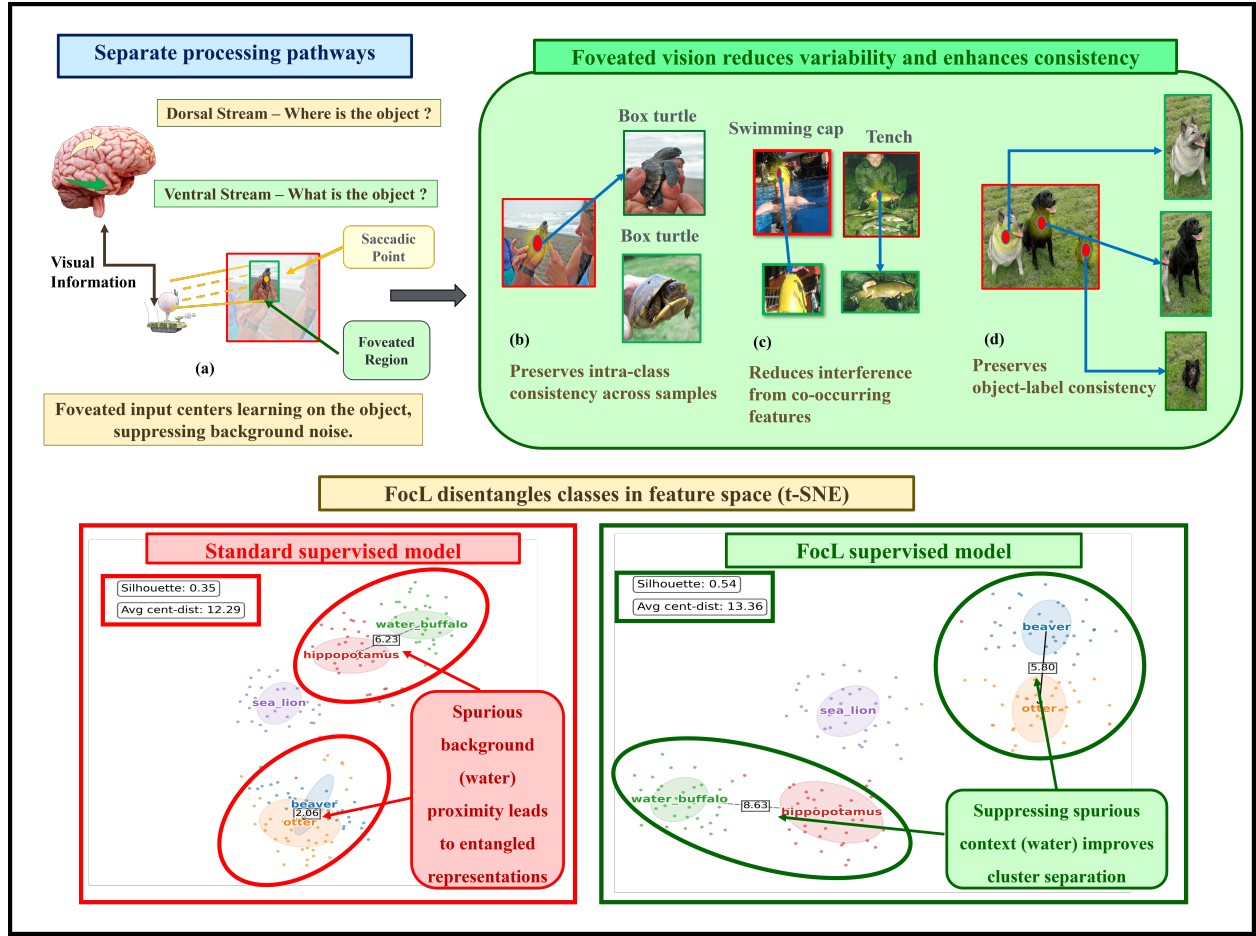

Figure 2: **FocL emulates human foveated vision to improve generalization by suppressing spurious context.** (a) FocL uses object-centric glimpses inspired by human visual streams to focus on relevant features. (b–d) This object-centric bias leads to more robust learning outcomes (intra-class consistency, reduced interference, object-label alignment). (Bottom) *t*-SNE: FocL (right) achieves better class separation (silhouette +0.19, avg. centroid dist. +1.07), unlike standard models (left) that confuse distinct classes (e.g., hippopotamus/water buffalo) by relying on shared water backgrounds.

Inspired by these principles of biological vision, we introduce FocL, which trains networks on foveated, object-centric crops that isolate the foreground to simplify learning and improve generalization. We emulate saccades by jittering bounding-box centers to generate multiple object-focused glimpses. By suppressing background clutter and isolating task-relevant regions, *FocL reduces sample complexity, shifting hard instances from the tail toward the mode of the difficulty distribution.* Rather than requiring models to learn from visually complex scenes, FocL restructures the input space to emphasize object–label consistency, reframing image classification as a simpler, more targeted task. This object-centric strategy improves generalization on foveated inputs and reduces memorization. FocL models require larger adversarial perturbations and converge faster, enabling learning with less data. Figure 2 visualizes this effect in a t-SNE projection, where FocL produces tighter semantic clusters. In contrast, a standard model entangles distinct classes, such as *hippopotamus* and *water buffalo*, by relying on shared background context (e.g., water). By focusing on foveated object regions, FocL learns cleaner semantic boundaries and more robust representations.

**Our contributions are as follows:**

- **FocL: Object-centric training strategy.** We introduce **FocL**, a training method that generates object-centric glimpses by jittering ground-truth boxes, encouraging models to focus on foreground features. (Sec 3)

- **Reduced memorization and more stable features.** FocL mitigates overfitting to non-robust contextual cues by restructuring visual input around object-centric crops. It reduces mean cumulative sample loss by approximately 65%, indicating a shift toward easier and more stable learning dynamics. Using Feldman–Zhang memorization scores intersected with our training split, FocL lowers cumulative sample loss on virtually all (>99%) of the high-memorization samples. FocL also increases the mean $\ell_2$ perturbation distance required to flip predictions by approximately 62%, reinforcing the connection between reduced memorization and more stable feature representations. (Sec 4.1)

- **Improved generalization on object features.** The reduction in memorization leads directly to stronger object-focused generalization. FocL improves Top-1 accuracy by up to 11% over baseline classifiers when each model is evaluated using its appropriate input modality. When paired with SAM as a dorsal proposal generator, FocL further provides an approximately 7 pp gain over baselines on a fixed validation subset considered under the **Any-correct** metric, which serves as an empirical performance ceiling for the Classifier+SAM system. (Sec 4.2)

- **Transfer to ImageNet-V2 and COCO.** Extending the dorsal–ventral setup, pairing SAM with FocL improves recognition under natural distribution shift on ImageNet-V2, yielding an approximately **8** pp gain over SAM paired with a standard classifier. Evaluated on a 1.5 K-image subset from COCO, the same combination achieves 3–4 pp higher mAP at IoU 0.3–0.5, demonstrating cross-dataset generalization in multi-object scenes despite no COCO-specific training. (Sec 4.2)

- **Enhanced learning dynamics and sample efficiency.** The simplified, object-centric supervision in FocL yields smoother optimization (approximately 46% lower mean gradient norms) and faster convergence, enabling comparable or higher accuracy with roughly 56% fewer training images under object localization (bounding boxes or proposals) than the standard baseline. (Sec 4.3)

Our focus is on systematically demonstrating that a foveated, object-centric input design yields consistent gains across learning dynamics, generalization, and cross-domain transfer. Our FocL classifier acts as a specialist module that enhances standard backbones and interfaces naturally with foundation models such as SAM. This demonstrates that biologically inspired design principles can strengthen conventional classifiers without modifying their architecture.

## 2 Related Work

We provide here a compact yet comprehensive survey of work most relevant to FocL; an expanded version is provided in Appendix A.1.

**Object-centric and foreground-focused learning.** Unsupervised methods such as MONet (Burgess et al., 2019) and Slot Attention (Locatello et al., 2020) aim to disentangle objects, whereas attention add-ons (e.g., CBAM (Woo et al., 2018)) and discovery pipelines like CutLER (Wang et al., 2023) modulate full-image features or mask foregrounds after the fact. A related thread learns *where* to look through iterative policies, exemplified by RANet (Mnih et al., 2014), Saccader-style models (Elsayed et al., 2019), GFNet (Wang et al., 2020), FABLE (Ibrayev et al., 2024a), and FALcon (Ibrayev et al., 2024b). **FocL instead trains on explicitly supervised foveated crops**, aligning the foreground with its label and suppressing background interference. Object localization can be delegated to external detectors (e.g., SAM (Kirillov et al., 2023; Ravi et al., 2024) or FALcon (Ibrayev et al., 2024b)) at inference, while our focus remains on the learning benefits of an object-first bias.

**Memorization in long-tailed learning.** Networks typically fit frequent patterns before memorizing rare, noisy, or atypical tail instances (Arpit et al., 2017; Feldman & Zhang, 2020). Theory and evidence suggest such memorization can sometimes be necessary for accuracy under skewed data (Brown et al., 2021; Usynin et al., 2024), yet it raises fairness, robustness, and privacy concerns (Li et al., 2025). Recent analyses propose proxies like Cumulative Sample Loss (CSL) (Ravikumar et al., 2025) and link high input-loss curvature to memorized long-tail samples (Garg et al., 2024; Ravikumar et al., 2024). While our work focuses on

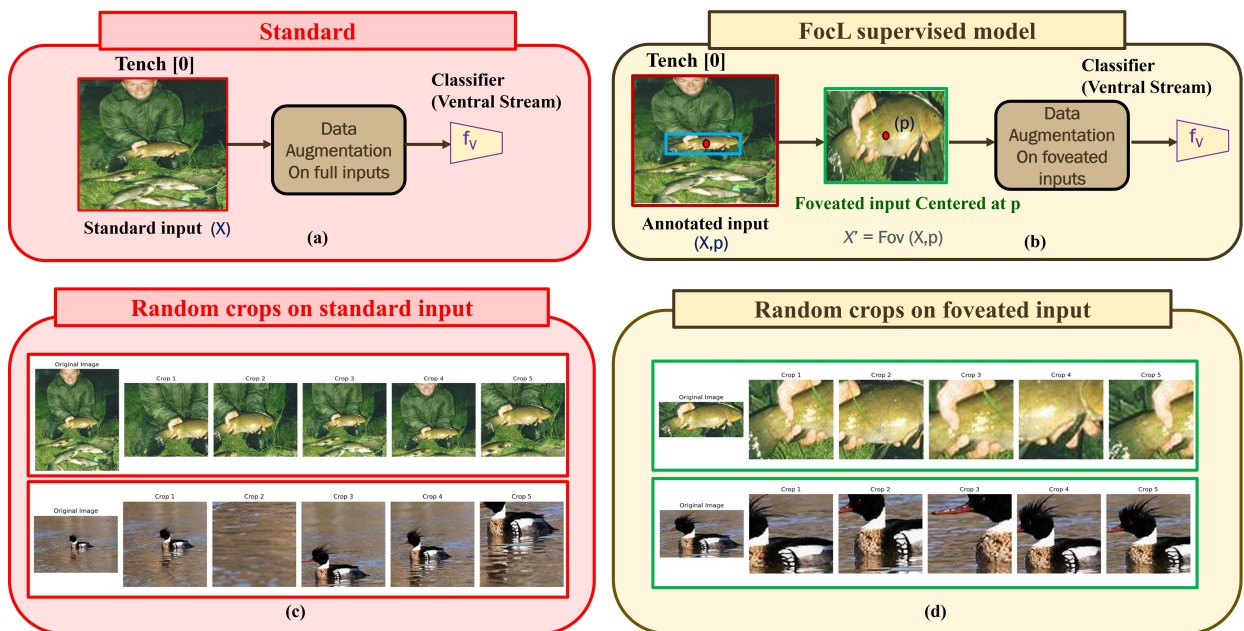

Figure 3: (a) Standard training uses the full image. (b) FocL replaces the raw input with foveated crop/crops centered on the annotated object. (c–d) Effect of Random-Resized-Crop augmentation under both pipelines. Each row shows the original image (left) followed by five crops seen across training epochs. In (c), full-image augmentation often captures irrelevant background (e.g., a fisherman's jacket or just water), encouraging spurious correlations. In contrast, (d) applies the same augmentations to foveated crops, yielding object-centric views that preserve foreground features. These cleaner views lead to more disentangled, object-aligned representations (see t-SNE, Figure 2).

supervised ImageNet settings, similar unintended "déjà-vu" memorization effects have been reported in self-supervised models (Meehan et al., 2023; Kokhlikyan et al., 2024) and vision-language models (Jayaraman et al., 2024). Complementary to studies analyzing these memorization dynamics, FocL restructures inputs to remove background clutter, simplifying hard examples and reducing reliance on brittle shortcut cues (Geirhos et al., 2020). While methods such as Mixup, CutMix, or logit adjustment (Zhang et al., 2018; Yun et al., 2019) mitigate class-level imbalance, FocL addresses instance-level difficulty directly.

**Foveation, robustness, and our contribution.** Recent robustness-oriented work blurs or down-samples the periphery, such as R-Blur for adversarial defense (Shah et al., 2023), textural encodings for IID gains (Gant et al., 2021), and active-vision systems that integrate multiple glimpses against transferable attacks (Mukherjee et al., 2025). These methods still retain substantial background context, and the robustness–memorization relationship remains delicate; for instance, adversarial training can induce robust overfitting (Dong et al., 2022). FocL adopts a different stance: it restricts the visual field to supervised object-centric crops, substantially reducing peripheral and background information while preserving label-consistent context. The observed increase in mean adversarial distance and the substantial drop in CSL are natural consequences of FocL simplifying each learning instance, rather than outcomes of explicit robustness optimization. This improved learnability is accompanied by smoother convergence and more stable, generalizable representations.

## 3 Methodology

In this section, we introduce **FocL**, our multi-glimpse foveated learning framework for visual recognition. We begin by briefly reviewing standard supervised learning and highlight how data augmentation behaves differently when applied to global versus foveated inputs. We then describe FocL in detail.

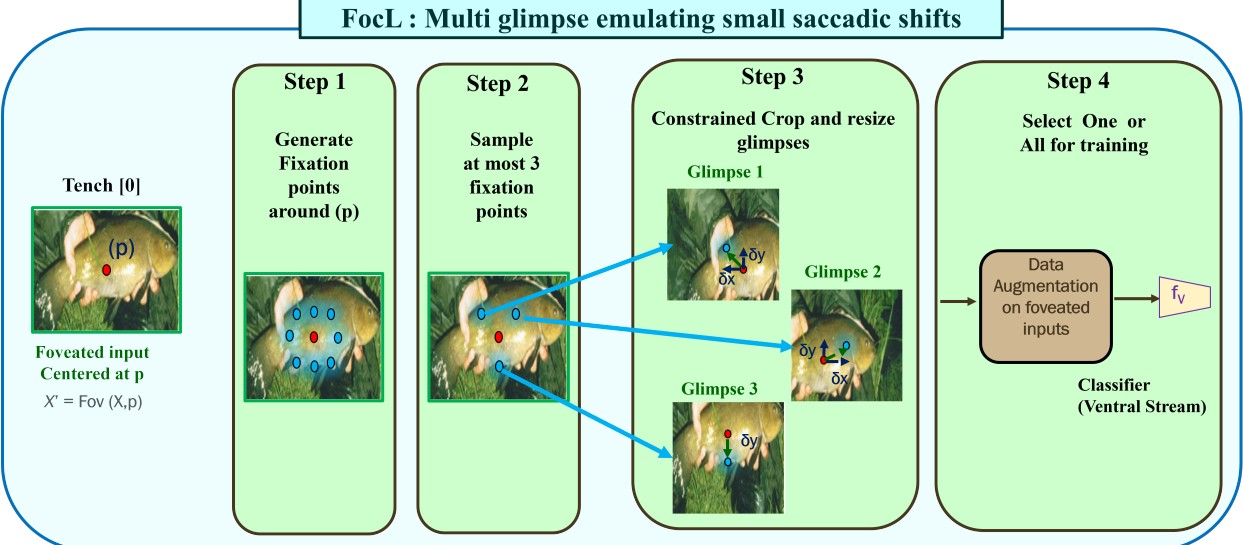

Figure 4: **FocL** with structured glimpse variation (Steps 1–4) simulates small saccadic shifts by jittering the fixation point and extracting up to three constrained crops around the object. Each glimpse is resized to the input resolution and used individually or jointly during training. These object-centric views reduce the influence of background clutter and encourage the network to focus on label-relevant foreground features, promoting stronger alignment between object structure and class semantics.

**Standard Supervised Training:** In conventional supervised pipelines (red panel, Figure 3a), the classifier $f$ is trained directly on full-resolution images using the standard cross-entropy loss:

$$\mathcal{L}_{\text{sup}} = \mathbb{E}_{(x,y)\sim\mathcal{D}} \left[ \ell(f(x),\, y) \right],$$

where $(x, y)$ denotes an image-label pair and $\ell$ is the classification objective. Since the entire image $x$ serves as input, data augmentations such as random resized cropping, horizontal flipping, and color jitter are applied globally across both foreground and background regions. This global augmentation strategy can introduce semantic drift (i.e., misalignment between label semantics and the visual content of augmented crops): the network may passively learn background features that are unrelated to the object label. As shown in Figure 3c, random crops observed during training emphasize irrelevant context, such as the fisherman's jacket instead of the tench (top row), or mostly water instead of the red-breasted merganser (bottom row). Such misaligned augmentations promote spurious correlations between background and label, causing the model to overfit to incidental context rather than learning object-centric, generalizable representations.

## 3.1 FocL: Foveated Object-Centric Learning

Given a labeled image $(x, y)$, we define the annotated bounding box as $b = (x_{\min}, y_{\min}, x_{\max}, y_{\max})$, and let its geometric center define a surrogate saccadic fixation point $p \in \mathbb{R}^2$. All pixels within $b$ are treated as the effective foreground region representing the object associated with label $y$; hence, the fixation is supervised and object-aligned. While biological foveation involves gradual spatial falloff and peripheral blur, we approximate it using a hard foveated glimpse obtained by cropping around $p$ to retain the labeled foreground and suppress most surrounding context. Because bounding boxes typically include some peripheral pixels, the glimpse may contain limited background, but remains substantially more object-focused than full-image crops. This setup is visualized in Figure 3b.

Using this formulation, we instantiate **FocL**, a strategy that generates multiple object-focused glimpses (up to three per image) by applying small, controlled spatial and scale jitter around the initial fixation point $p$ ($\pm 5$–$10\%$ of box width/height and $\pm 5\%$ scale range; details in Appendix). These jittered glimpses serve to relax tight bounding boxes, emulate human-like saccadic sequences, introduce mild viewpoint variations, and mitigate geometric distortions from resizing. By primarily exposing the model to these varied object-centric views, **FocL** encourages a strong inductive bias towards foreground features over background clutter.

Consequently, even when standard augmentations are applied, these glimpses maintain semantic consistency and preserve object identity (yellow panel, Fig. 3d).

For each image, we extract up to $k = 3$ foveated glimpses (a tunable parameter) and treat them as individual training examples that share the same label. During training, these glimpses are included in the same mini-batch (i.e., not shuffled across images) to preserve intra-object coherence while still benefiting from batch diversity. This setup enables the model to jointly process multiple views of the same object, reinforcing consistent foreground–label mappings. The per-sample loss is computed by averaging over all glimpses as follows:

$$\mathcal{L}_{\text{FocL}} = \mathbb{E}_{(x,y)\sim\mathcal{D}} \left[ \frac{1}{k} \sum_{i=1}^{k} \ell\big(f(\text{Fov}_i(x, p_i)), y\big) \right].$$

Here, $\text{Fov}_i(x, p_i)$ denotes the $i^{\text{th}}$ foveated crop generated around a distinct jittered fixation point $p_i$, sampled from a small neighborhood of the annotated object center $p$. Although each crop uses its own offset $p_i$, we write $\text{Fov}_i(x, p)$ to indicate that all glimpses are relative to the same base annotation. The generation of these object-focused glimpses (illustrated in Figure 4) first samples fixation centers within a small neighborhood around the annotated point. Candidate centers are then adjusted to respect image boundaries, followed by distortion-aware cropping that preserves object aspect ratio during resizing. This design encourages robust learning primarily from foreground regions while maintaining mild variability in position and scale. A detailed algorithm, including jitter magnitude and selection criteria, is provided in Appendix A.2. We apply BatchNorm in the standard way over the concatenated crop batch (across all images and glimpses); while crops from the same image are correlated, each mini-batch still contains many distinct images and independently augmented crops, so correlation mainly enforces intra-object coherence rather than dominating BN statistics.

## 4 Experiments

We evaluate FocL across three dimensions: **robustness to memorization**, **generalization under foveated inputs**, and **sample efficiency**. Details regarding hyperparameters and reproducibility are provided in Appendix A.3.

### 4.1 Does FocL reduce memorization?

We address this question through two complementary analyses. First, we examine *Cumulative Sample Loss* (CSL) introduced by Ravikumar et al. (2025), a proxy for sample-level learning difficulty, to quantify how FocL reshapes training dynamics. The CSL distribution shifts from the tail toward the mode, indicating that previously hard examples become easier to learn. To relate this effect to memorization, we align CSL values with pre-computed scores from Feldman & Zhang (2020) for our training set's ImageNet indices. By identifying the top 1% most memorized samples, we isolate a cohort of 820 verifiably memorized examples. FocL is exceptionally effective for this group, making 99.88% of these samples easier to learn (lower CSL). Finally, we evaluate adversarial resistance to assess whether this reduction in memorization also yields more stable and robust features.

**Setup** All memorization analyses are conducted on the 100K ImageNet partition (details in Appendix A.3), using 85K samples for training, 15K for validation. FocL is evaluated in its single-glimpse configuration ($k=1$), while the standard model is trained on full images. Both use identical hyperparameters and training protocols, ensuring that any observed differences arise solely from the input representation.

**Learning Difficulty and Memorization.** To investigate how FocL mitigates memorization, we analyze CSL, a proxy for learning difficulty where higher values correspond to harder-to-learn samples. On an aggregate level, FocL substantially reshapes the CSL distribution (Fig. 5, left), reducing the mean CSL from 206.66 to 72.30. To understand the source of this improvement, we conduct a targeted analysis using pre-computed memorization scores from Feldman & Zhang (2020). By intersecting the indices of the top

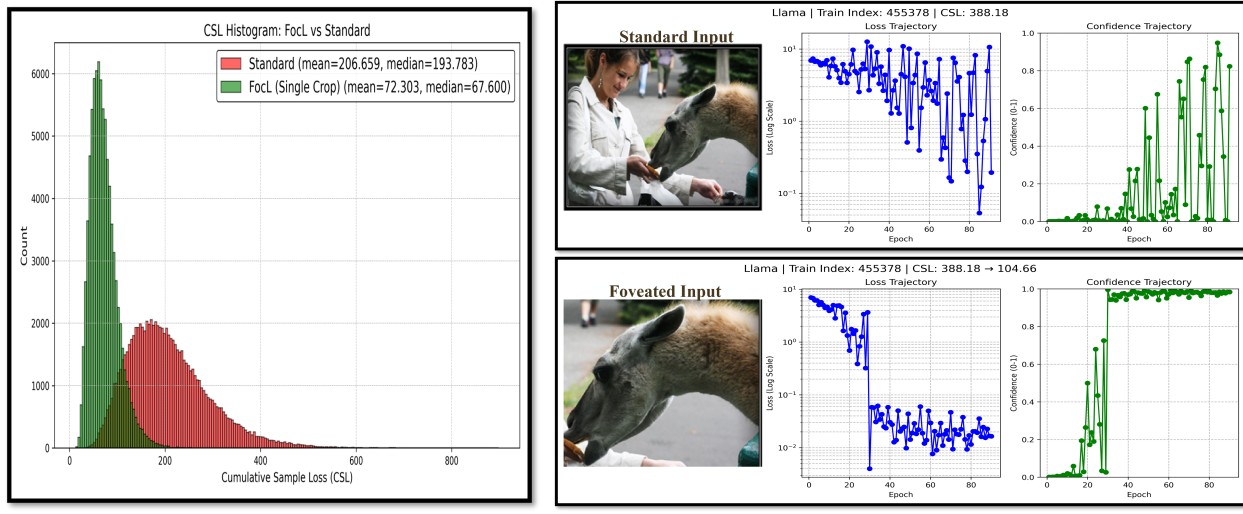

(a) Distribution over entire training set      (b) Loss and confidence trajectories for a sample

Figure 5: Cumulative sample loss (CSL) memorization proxy analysis. **Left:** Shift from tail to mode. FocL exhibits significantly lower mean and median CSL, and the distribution is tightly concentrated toward lower values, indicating that samples become easier to learn due to object-centric inputs and reduced contextual interference. **Right:** Example of a high-CSL sample from the Llama class. In the standard model, background elements like the human introduce semantic contamination, leading to noisy loss and confidence trajectories. With FocL, foveated input enables more stable learning, reflected in the smoother trajectories and a large CSL drop from 388.18 to 104.66.

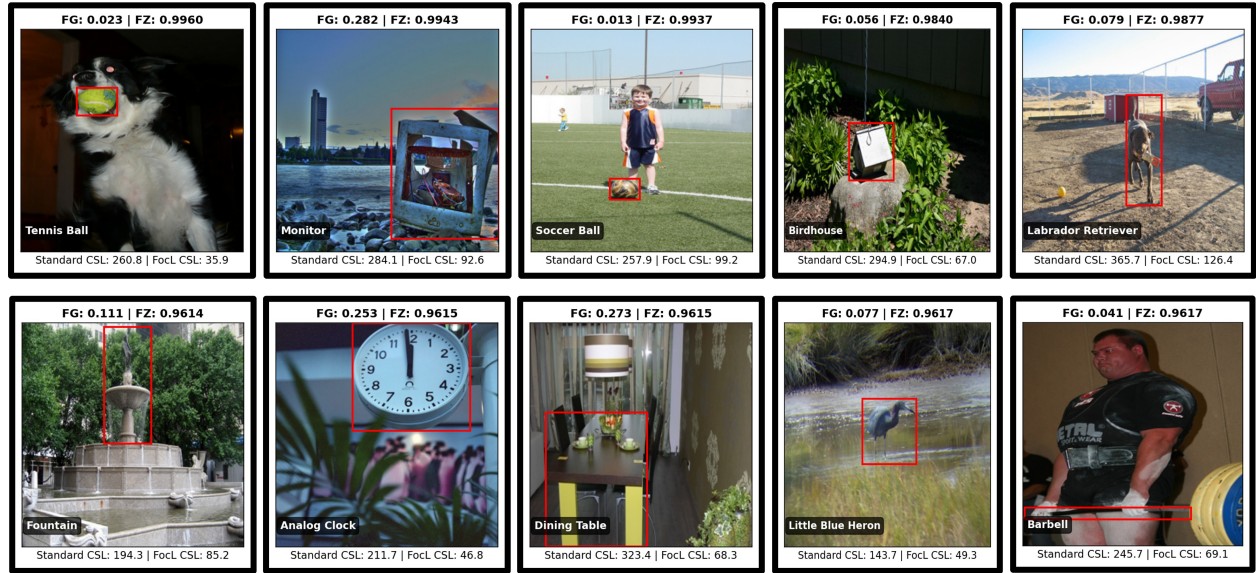

Figure 6: **Example visualizations from the top 1% most memorized ImageNet samples (Feldman & Zhang scores).** We show representative examples that often include surrounding context beyond the target object; FG is reported only descriptively in these figures (box-area / image-area) (e.g., the 'Soccer Ball' example has FG = 0.013). By isolating the object region, FocL reduces learning difficulty, reflected by large per-sample drops in **CSL**.

1% most memorized ImageNet samples with our 85K training set, we identify a cohort of 820 verifiably memorized examples. For this group, FocL is exceptionally effective, making 99.88% of these samples easier to learn ($p < 0.001$). We assess significance with a one-sided paired sign (exact binomial) test over the 820 samples, where a sample is counted as improved if $\text{CSL}_{\text{FocL}} < \text{CSL}_{\text{Std}}$; full hypotheses and the exact

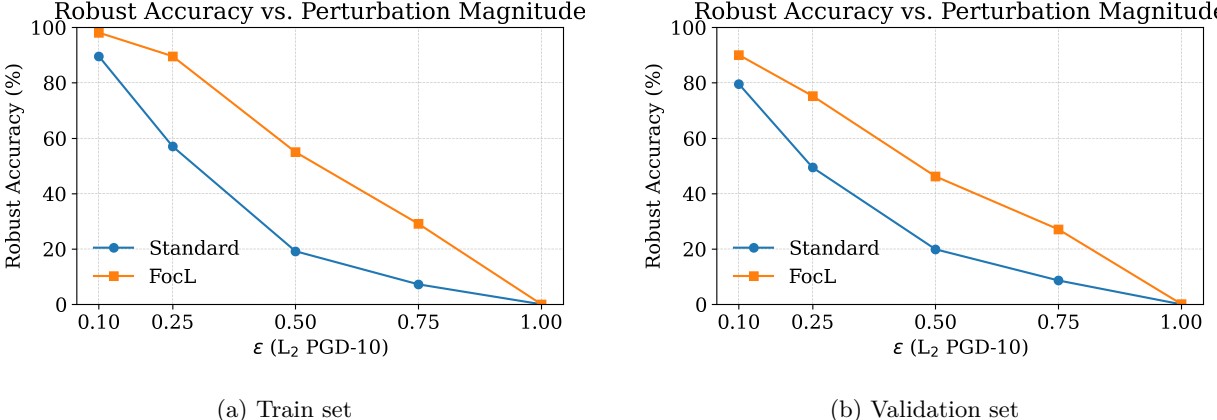

(a) Train set

(b) Validation set

Figure 7: Robust accuracy of standard and FocL models against $\ell_2$ PGD-10 perturbations. At $\epsilon = 0.25$, FocL achieves 89.59% on the training set versus 57.05% for the standard model, a gap of 32.54 percentage points. On the validation set, the gap is 25.82 percentage points (75.24% vs. 49.42%). The standard model's accuracy drops sharply at small perturbation levels ($\epsilon < 0.25$), with many predictions flipped by minimal adversarial budgets. This behavior, especially evident on the training set, suggests memorization and a reliance on brittle and non-generalizable features. The degradation is less pronounced in FocL, likely due to its robust object-centric feature learning that reduces background interference.

$p$-value are reported in Appendix A.6. We additionally include example visualizations from the top-1% FZ set (Figs. 6 and 12) as qualitative examples of memorized samples; in these figures, we report FG as the ground-truth bounding-box area divided by the full image area. Overall, this targeted analysis shows that FocL's aggregate gains are driven by reducing learning difficulty on such problematic instances, shifting mass from the tail of the difficulty distribution toward its mode.

**Adversarial Resistance**  To further probe the stability of learned representations rather than explicit adversarial robustness, we apply projected-gradient-descent (PGD) $\ell_2$ perturbations (Madry et al., 2018) on a balanced ImageNet subset. For fairness, each model is attacked on its native input type—full images for the standard baseline and foveated crops for FocL. The analysis indicates that FocL-trained representations are harder to perturb, reflecting increased stability. The mean adversarial distance, defined as the average $\ell_2$ perturbation required to alter a model's prediction, increases by **62%** under FocL training ($\bar{d} = 0.6169$ vs. 0.3806 for the standard model), showing that substantially stronger perturbations are needed to induce misclassification. Moreover, FocL maintains consistently higher accuracy across increasing perturbation magnitudes ($\epsilon$), as shown in Figure 7. In contrast, the standard model exhibits a steep accuracy drop even at small $\epsilon$, suggesting that it relies on brittle, non-generalizable features often linked to memorization (Carlini et al., 2019). This evidence complements the CSL analysis, reinforcing that FocL reduces memorization by promoting more stable and semantically aligned features.

**Key Takeaways.**  FocL primarily reduces memorization by simplifying learning at the *sample level*. It markedly lowers mean Cumulative Sample Loss, shifting hard examples from the tail toward the mode of the difficulty distribution, and alleviates background-driven memorization observed in the most challenging samples. This improvement in learnability directly translates into greater feature stability: when attacking each method on its native input to match its training regime (both use 224×224 inputs: full images for the standard baseline and foveated crops for FocL; no resizing within the attack loop), FocL requires significantly larger $\ell_2$ perturbations to flip predictions. We note that cropping changes the input content distribution (e.g., reduced background clutter), which can make perturbations harder to "hide" in background regions; accordingly, these distances should be interpreted as stability of the deployed end-to-end object-centric input representation plus classifier, consistent with more robust, object-centered features. This is an intended property of foveated perception, since the goal is to suppress background-driven cues and focus decisions on object-relevant evidence. Additional results are provided in Appendix A.6, and A.5.

Table 1: **Full-image, Oracle, and active localization (FALcon) performance on ImageNet-V1 (2 K subset).** Top-1 / Top-5 accuracy (%) for Standard and FocL classifiers under full-image input, oracle (GT) crops, and FALcon multi-glimpse aggregation.

| Single-glimpse setting | Standard (Full Image) | Standard (Oracle) | FocL (Oracle) |
|---|---|---|---|
| Top-1 (%) | 60.90 ± 1.10 | 59.83 ± 1.35 | **72.23** ± 0.66 |
| Top-5 (%) | 83.05 ± 0.74 | 82.05 ± 0.48 | **90.82** ± 0.42 |
| **Multi-glimpse (FALcon)** | **Avg** | **Voting** | **Weighted** |
| Standard (Top-1) | 61.25 ± 0.47 | 60.10 ± 0.92 | 60.52 ± 0.94 |
| FocL (Top-1) | **61.45** ± 0.11 | **60.68** ± 0.37 | **61.37** ± 0.24 |

### 4.2 Does FocL improve generalization under foveated inputs?

**Experimental Setup.** We use the ImageNet subset with bounding box annotations similar to Deng et al. (2009); Russakovsky et al. (2015), containing 482K images. All models adopt the standard ResNet50 backbone (He et al., 2015); the FocL variant employs foveated glimpse(s) as described in Sec. 3 (single-crop or multi-crop, as specified per experiment). FocL and standard models are trained with identical settings to ensure a fair comparison. Our classifier does not take segmentation as an additional input channel; ground-truth boxes (masks when used), are used only to *define* object-centric inputs (crop or masked image), and the ResNet-50 backbone architecture is unchanged.

We evaluate generalization under three complementary settings: (1) *oracle bounding box evaluation* using a single ground truth crop per image, (2) *multi glimpse aggregation* with FALcon introduced by Ibrayev et al. (2024b), and (3) an *empirical high performance dorsal ventral evaluation* using Segment Anything (SAM) developed by Ravi et al. (2024). For the first setting, each model is evaluated in its respective input domain, using full images for the standard classifier and bounding box crops for FocL. In addition, the standard model is also tested on oracle crops as a diagnostic to assess its reliance on background context. For FALcon and SAM, both models receive identical localized crops from the respective localizer, enabling a fair comparison. All ImageNet V1 experiments, including oracle and FALcon evaluations, use the same 2,000 validation images for a fair comparison. For shifted distribution testing, we evaluate on 2,000 images from the ImageNet V2 (MatchedFrequency) set. **To further isolate whether the gains arise merely from background/context removal or from our foveated training, we provide additional intermediate baselines (HardMask/MeanPad and RoIAlign) in Appendix A.10-13.** Finally, we extend this framework to a cross-domain generalization on COCO, where SAM and both classifiers are tested without any COCO specific training.

**Oracle Bounding Box Evaluation.** We first evaluate performance under ideal foveation using oracle bounding box crops (Table 1-first two rows). Each model is tested in its native domain, with the standard model evaluated on full images and FocL on its corresponding foveated crops. To assess context dependence, the standard model is also tested on oracle crops as a diagnostic. Its Top 1 accuracy drops by 1.07 pp, indicating reliance on background cues. When compared across their respective inputs, FocL outperforms the standard model by **11.33 pp** in Top 1 and **7.77 pp** in Top 5 accuracy, highlighting its stronger generalization under object centered supervision. Additional ablations are provided in Appendix A.4. Please refer to Appendix A.12 (Table 11), where we compare against HardMask and RoIAlign under oracle crops and observe that FocL's advantage prevails over both baselines. This highlights the benefit of FocL's glimpse-based, jittered object-aligned training beyond simple mean-padding based context suppression or RoI-style feature pooling.

**Multi Glimpse Aggregation with FALcon.** We next evaluate both models within an active vision setup using FALcon, which provides object centric glimpses for classification (Table 1, bottom rows). Unlike the oracle setting, FALcon introduces viewpoint changes and uses automatically localized bounding boxes that are not always precise, leading to a small drop from ideal foveation but offering a more practical scenario. The standard model shows minor variation across aggregation schemes, achieving 61.25%, 60.10%,

Table 2: **Generalization under foveated inputs on ImageNet-V1 (in-distribution) and ImageNet-V2 (out-of-distribution) subsets.** Top-1 accuracy (%) for three inference pipelines: Standard classifier only, SAM + Standard classifier, and SAM + FocL classifier. FocL consistently outperforms the standard model across both domains, while the dorsal like SAM proposals illustrate the high performance potential of the active vision system.

| Dataset | Standard only | SAM + Standard | SAM + FocL |
|---|---|---|---|
| ImageNet-V1 | $60.90 \pm 1.10$ | $68.08 \pm 0.97$ | **75.82** $\pm 0.23$ |
| ImageNet-V2 | $49.77 \pm 0.94$ | $57.50 \pm 1.92$ | **65.80** $\pm 0.55$ |

Table 3: Cross-domain generalization setup on COCO. mAP at IoU = 0.3 and 0.5 for two SAM proposal regimes. Values show performance improvement of the overall SAM + FocL system relative to the SAM + Standard baseline.

(a) Max 20 proposals / image

| IoU | SAM + Standard → SAM + FocL |
|---|---|
| 0.3 | $24.39 \rightarrow$ **28.22** (+3.83 pp / +16%) |
| 0.5 | $13.10 \rightarrow$ **16.22** (+3.12 pp / +24%) |

(b) Max 300 proposals / image

| IoU | SAM + Standard → SAM + FocL |
|---|---|
| 0.3 | $28.19 \rightarrow$ **31.76** (+3.57 pp / +12.7%) |
| 0.5 | $14.29 \rightarrow$ **17.24** (+2.95 pp / +20.6%) |

and 60.52% Top 1 accuracy for Average, Voting, and Weighted aggregation respectively. FocL maintains consistently higher performance with 61.45%, 60.68%, and 61.37% Top 1 accuracy, demonstrating improved stability and better generalization under multi glimpse viewpoint variations.

**Evaluation with SAM as a Dorsal Localizer.** We next integrate the classifiers with the Segment Anything Model (SAM) developed by Ravi et al. (2024), whose hyperparameters are provided in the Appendix. SAM serves as a strong dorsal module that can segment and generate object proposals with high precision across a wide range of object sizes. This expressive foundation model complements our specialist FocL classifier, together forming a dorsal ventral system that offers a high performance reference for evaluating classification accuracy under the relaxed Any-Correct criterion. As shown in Table 2, SAM significantly improves performance for both classifiers. On ImageNet V1, pairing SAM with the standard classifier yields 68.08% Top 1 accuracy, while the SAM and FocL combination achieves 75.82%, marking a 7.7 pp gain. This advantage further increases on ImageNet V2 (Recht et al., 2019), where SAM and FocL reach 65.80%, outperforming the baseline by more than 8 pp. While the monolithic standard classifier tends to capture spurious background statistics specific to ImageNet V1, FocL learns object aligned representations that generalize effectively to out of distribution data, yielding more stable behavior under natural shifts. Feature-pooling baselines such as RoIAlign exhibit a similar pattern under SAM proposals, performing competitively on ImageNet-V1 but dropping more sharply on ImageNet-V2, as shown in Table 12. Appendix A.13 provides a much more detailed comparative analysis. Details regarding SAM hyperparameters and setup are provided in Appendix A.4.D and A.4.E.

**cross-domain generalization from ImageNet to COCO.** To examine cross-domain generalization, we evaluate the SAM and classifier pipelines on the COCO dataset without any fine tuning. This setup is not intended as a competitive benchmark but as a diagnostic test of transferability—isolating whether the model has learned robust object features or merely dataset-specific background statistics—consistent with prior studies that evaluate ImageNet trained models on COCO style tasks (He et al., 2018; Shin et al., 2023). Public frameworks such as *ImageNet-to-COCO* by Lo (2020) similarly map overlapping ImageNet and COCO categories to enable cross-domain analysis. We follow this mapping, where 65 of the 80 COCO categories align directly with ImageNet classes, and use 1.5K images from the COCO training set for evaluation since these samples are generally more cluttered and challenging than validation images. SAM serves as the dorsal localizer, producing up to 50 proposals per image, while the ImageNet trained Standard and FocL classifiers predict object labels for each proposal. Mean average precision (mAP) is computed at IoU thresholds of

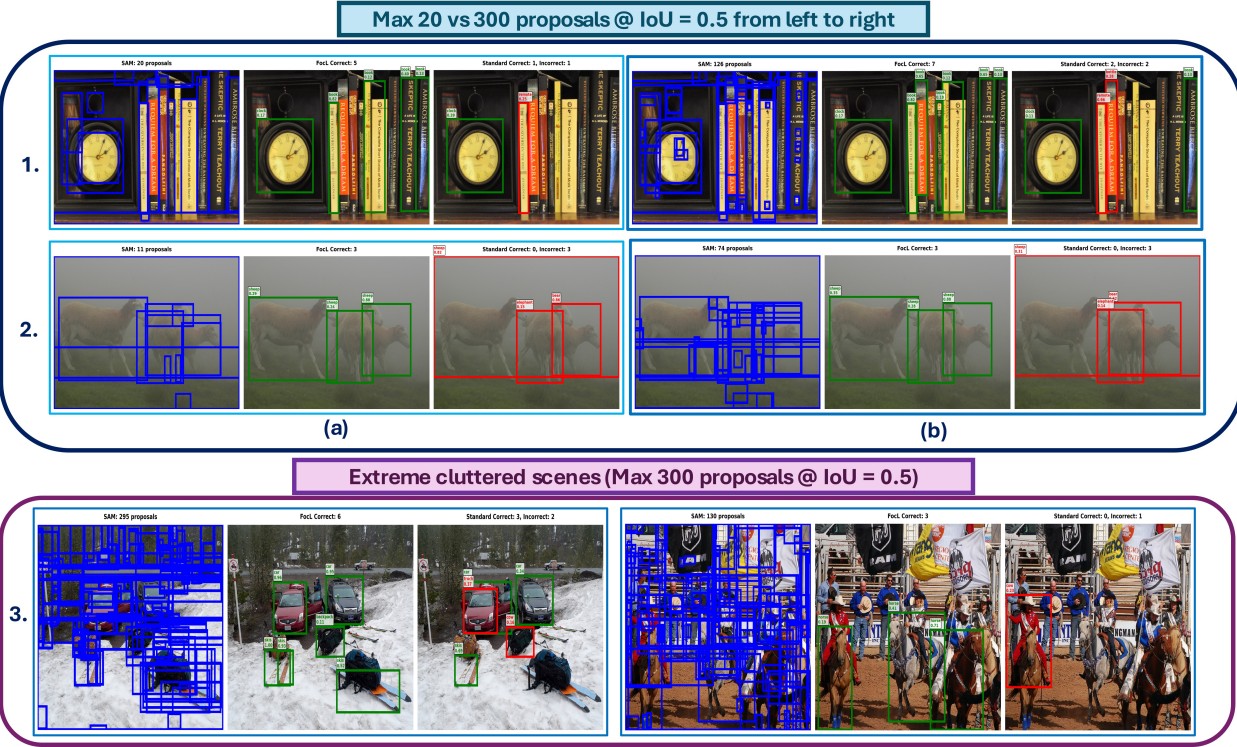

Figure 8: **Cross-domain generalization in a zero-shot transfer setup (IoU = 0.5).** SAM (blue) provides region proposals to the ventral classifiers (FocL + SAM in green, Standard + SAM in red). Rows 1–2 show identical COCO images under two proposal regimes: (a) 20 proposals and (b) 300 proposals per image. FocL maintains stable object-centered detections and suppresses background confusion as proposal density increases. Row 3 depicts extreme clutter (300 proposals), where FocL yields tighter and semantically consistent predictions while the Standard model mislabels overlapping objects, demonstrating improved cross-domain transfer without any COCO-specific training.

0.3 and 0.5. We include the relaxed IoU=0.3 threshold to account for potential localization drift when transferring to an unseen domain.

As shown in Table 3, the SAM and FocL pipeline consistently outperforms the SAM and Standard setup across IoU thresholds and proposal counts. With 20 proposals per image, FocL improves mAP by 3.83 pp at IoU 0.3 and 3.12 pp at IoU 0.5 (16% and 24% relative gains). Increasing the proposals to 300 further raises performance to 31.76 and 17.24, respectively. Qualitative examples in Fig. 8 illustrate cleaner, object centered detections and fewer spurious boxes. In highly cluttered scenes (Row 3, second image), the SAM and FocL system correctly identifies three distinct objects, while the SAM and Standard combination misses all, highlighting FocL's stronger object awareness and cross-domain robustness. More details are provided in Appendix A.4 (C, F, G, H).

### 4.3 Does FocL enhance learning efficiency and stability?

To enable fair comparisons, experiments in this section use the FocL single-crop variant against standard full-image training. We assess FocL's impact on optimization stability, and data efficiency. Unless otherwise stated, all results in Sec. 4.3 compare **FocL (single-crop)** against standard full-image training using an identical, fixed ImageNet training recipe (no method-specific hyperparameter tuning or setting-specific schedule adjustments). Training details are provided in Appendix A.3.

**Smoother Optimization via Gradient Norm Reduction.** Analysis of $\ell_2$ gradient norms during training on the 85K subset further reveals FocL's stabilizing effect. FocL exhibited consistently lower gradient

Table 4: **Comparison across dataset scales and FocL variants on ImageNet.** Top-1 / Top-5 accuracy (%) for Standard ResNet-50 trained on the full 1.03M ImageNet set and the 482K subset, and for FocL single-crop (SC) and multi-crop (MC) models trained on the same 482K subset.

| Model Type | 1.03M Standard | | 482K Standard | | FocL SC | FocL MC |
|---|---|---|---|---|---|---|
| | Full Img | BBox | Full Img | BBox | BBox | BBox |
| Top-1 | 70.65 | 67.10 | 60.90 | 59.83 | 71.05 | **72.23** |
| Top-5 | 89.35 | 85.70 | 83.05 | 82.05 | 89.80 | **90.82** |

magnitudes. Specifically, the mean gradient norm per parameter (normalized by model size) was reduced by approximately 45.8% (from $1.49 \times 10^{-3}$ for standard to $8.08 \times 10^{-4}$ for FocL). This substantial drop suggests FocL creates a simpler optimization landscape with less gradient noise. (Absolute mean gradient norms: Standard $3.81 \times 10^4$, FocL $2.07 \times 10^4$)

**Sample Efficient Learning.** FocL's simplified learning paradigm leads to strong sample efficiency (Table 4). When trained on the 482K annotated ImageNet subset, the FocL multi crop model achieves 72.23% Top 1 and 90.82% Top 5 accuracy when evaluated using oracle bounding boxes. This performance surpasses the standard model trained on an 80% partition of the full 1.28M ImageNet dataset ($\approx$1.02M images), which attains 70.65% and 89.35% Top 1 and Top 5 accuracy respectively, despite using more than twice the data. These results show that FocL achieves competitive generalization with significantly fewer training samples. *We emphasize that this efficiency is measured in the number of training images conditional on access to object localization (bounding boxes or proposals), and does not claim reduced human annotation cost relative to image-level labels. In practice, such localization can come from existing metadata or automated proposal methods (e.g., SAM), as studied in our active-vision pipeline.* Importantly, FocL is not a coreset or data pruning method; its efficiency gains arise from foreground aligned input restructuring rather than selecting or discarding samples. Additional analyses are provided in the Appendix A.8.

## 5 Conclusion

In this paper, we introduced FocL, a multi-glimpse training strategy that encourages models to learn object-centric features by reducing background clutter. By directing learning toward label-consistent object regions, FocL reduces memorization and yields more stable representations. A reduction of approximately 65% in mean cumulative sample loss and an approximately 62% increase in the $\ell_2$ perturbation required to flip predictions both indicate that FocL learns less brittle and more generalizable features. Under ideal foveated crops, FocL improves Top-1 accuracy by up to 11 pp compared to standard and intermediate baselines, showing that minimizing background context simplifies learning. When paired with SAM as a dorsal proposal generator, FocL further improves performance by about 2-7 pp on ImageNet-V1 and about 6-8 pp under natural distribution shift to ImageNet-V2. This object-centric bias also transfers across domains: on COCO, FocL yields over 12% relative gains in cross-domain transfer without any target-domain training. Finally, FocL reaches comparable or higher accuracy using roughly 56% less training data while maintaining smoother gradients, reflecting more efficient and stable optimization. Overall, foveated learning provides a simple and biologically grounded path toward models that memorize less, generalize better, and transfer more effectively across visual domains.

**Acknowledgments**

This work was supported in part by, the Center for the Co-Design of Cognitive Systems (CoCoSys), a DARPA-sponsored JUMP 2.0 center, the Semiconductor Research Corporation (SRC), and the National Science Foundation.

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

# A Appendix

## A.1 Expanded Related Work

This section expands on the related work most relevant to FocL, organized into three main areas: (1) object-centric and foreground-focused learning, (2) memorization and generalization in long-tailed settings, and (3) foveation-based methods for robustness and efficient learning.

**Object-Centric and Foreground-Focused Learning** Unsupervised object-centric models such as MONet (Burgess et al., 2019) and Slot Attention (Locatello et al., 2020) aim to decompose scenes into discrete object representations, but often struggle on complex natural images. Attention modules like CBAM (Woo et al., 2018) reweight spatial and channel-wise features post hoc, while pipelines like CutLER (Wang et al., 2023) attempt to discover and mask foregrounds, still operating over full-image inputs. A related class of models learns dynamic visual attention through iterative glimpses. RANet (Mnih et al., 2014) uses a recurrent attention network to focus on different image regions over time, while Saccader (Elsayed et al., 2019) and GFNet (Wang et al., 2020) emulate saccadic movements and process glimpses within computational budgets. FABLE (Ibrayev et al., 2024a) models a dorsal-ventral system using reinforcement learning to locate objects, and FALcon (Ibrayev et al., 2024b) further introduces saccades and foveation, enabling active multi-object detection even from single-object training. These models mimic human vision by sequentially sampling high-resolution glimpses, discarding background via task-adaptive attention. **FocL adopts a different paradigm. Rather than learning fixation policies, it uses supervised bounding boxes to directly crop foreground objects, fully removing background prior to training. This object-label alignment reduces contextual bias and simplifies training, focusing on the impact of this transformation on generalization, memorization, and convergence.**

**Memorization in Long-Tailed Learning**  Deep networks tend to memorize rare, noisy, or atypical examples after first fitting frequent and simpler patterns (Arpit et al., 2017; Feldman & Zhang, 2020). Arpit et al. (2017) show that during training, networks prioritize learning generalizable patterns but eventually begin memorizing outliers and noisy data. Feldman & Zhang (2020) further argue that memorization is not just incidental but sometimes essential for accurate predictions on tail samples, especially when such examples are underrepresented or conflict with dominant patterns in the data. Building on this, Brown et al. (2021) provide theoretical insights into why high-accuracy learners may be forced to memorize substantial information about training data in natural, long-tailed settings. Usynin et al. (2024) offer a comprehensive survey of memorization across multiple regimes, categorizing its benefits and drawbacks with respect to generalization and privacy. Li et al. (2025) take a systems-level view, framing memorization as central to the trustworthiness of machine learning systems. They explore its role across fairness, robustness, and data privacy, and propose a taxonomy to reason about these interactions based on data granularity such as class imbalance, noise, and atypicality. To characterize memorization quantitatively, Ravikumar et al. (2025) introduce the Cumulative Sample Loss (CSL), which tracks the cumulative training loss per sample. They show that hard-to-learn and noisy samples consistently exhibit higher CSL, providing a strong signal of memorization. Complementary to this, Garg et al. (2024) and Ravikumar et al. (2024) explore the curvature of the loss surface. Their results show that memorized examples lie in sharper regions of the landscape—i.e., with higher curvature. This often indicate less robust generalization and more brittle learning dynamics. Recent studies also demonstrate that memorization is not limited to supervised learning. Meehan et al. (2023) uncover "déjà vu" memorization in self-supervised models, where training samples are memorized even without explicit labels. Kokhlikyan et al. (2024) refine the measurement of this phenomenon, offering efficient evaluation tools for memorization in large SSL models. Similar memorization behavior is observed in vision-language models (Jayaraman et al., 2024), where individual image or object information is retained by the model even beyond its intended abstraction level. **FocL offers an input-level simplification by suppressing background clutter entirely, reducing reliance on spurious correlations and shortcut cues (Geirhos et al., 2020).** By restructuring the input itself, FocL shifts the learning task to focus on object-relevant features from the outset. Unlike techniques such as Mixup (Zhang et al., 2018), CutMix (Yun et al., 2019), or logit-adjustment methods, which alter training dynamics via label smoothing, augmentation, or reweighting, FocL tackles instance-level difficulty directly by improving input-label consistency through foveated, object-aligned supervision.

**Foveation, Robustness, and Efficient Learning**  Foveation-inspired methods have been explored as mechanisms for improving robustness. R-Blur (Shah et al., 2023) applies adaptive Gaussian blurring to simulate peripheral vision, improving resistance to adversarial attacks. Gant et al. (2021) use a Foveated Texture Transform to enhance both IID generalization and robustness. Active-vision systems (Mukherjee et al., 2025) formulate a deep learning-based dorsal-ventral architecture by building on prior works such as FALcon (Ibrayev et al., 2024b) and GFNet (Wang et al., 2020), and demonstrate improved robustness in black-box transfer attack scenarios. By processing sequential glimpses at multiple fixation points, the approach enhances adversarial resilience for both CNNs and transformer-based ventral networks, particularly under natural and transferable adversarial inputs. Luo et al. (2015) apply CNNs to foveated regions, achieving strong robustness to perturbations. R-Warp (Vuyyuru et al., 2020) and VOneBlock (Dapello et al., 2020) embed cortical and retinal processing into CNNs. Harrington & Deza (2022) show how robust models align with texture-based peripheral vision, and Shah et al. (2023) simulate peripheral degradation for robustness gains. **FocL introduces a simplified mechanism: a complete background cut-off via supervised crops. This restructuring results in cleaner, more learnable samples and exhibits a coupled effect; higher adversarial perturbation energy required to flip predictions and lower Cumulative Sample Loss (CSL). Both serving as indicators of reduced memorization.**

**Saliency guided data augmentation methods to reduce background**  Saliency-guided data augmentation provides a simple, automated way to reduce *background dependence* by preferentially preserving or mixing foreground-discriminative regions, thereby discouraging reliance on spurious background cues rather than sampling cut-and-paste areas uniformly at random. For example, Uddin et al. (2021) uses a saliency map to select an informative patch and paste it onto another image, mixing labels based on the pasted region. Similarly, Kim et al. (2020) exploits saliency and local image statistics to optimize the mixing

mask, producing more semantically meaningful composites for mixup-style training. Along the same line, Huang et al. (2021) leverages class activation maps to estimate semantic composition and set mixing proportions, reducing label noise in mixed samples (especially in fine-grained recognition). Gong et al. (2021) propose *KeepAugment*, which uses saliency maps to identify informative regions and then constrains standard augmentations to *preserve* those regions (e.g., avoiding cutting salient areas or pasting them back after transformation). Aniraj et al. (2023) study background-bias mitigation via masking, comparing *early masking* (removing background at the input image level) and *late masking* (masking background-corresponding spatial features), and show these strategies improve generalization under OOD background shifts. In contrast, FocL uses bounding-box–driven *foveated jitter* to enforce consistent, label-aligned object coverage (i.e., stronger supervision than heuristic saliency) and explicitly evaluates reduced *memorization*. It further extends to an active-vision pipeline where SAM acts as an external proposal network to generate foveated glimpses and demonstrate cross-dataset generalization.

FocL thus bridges perceptual inspiration with practical gains in generalization, memorization reduction, and efficient learning without requiring specialized architectures or costly training procedures.

### A.2 FocL glimpse generation algorithm details

**Overview.** The FOCL framework generates up to three object-centric glimpses per image, centered around a supervised fixation point derived from the annotated bounding box. These glimpses simulate small saccadic shifts near the object and reduce background clutter while preserving semantic alignment with the label. While the main paper outlines the high-level steps Figure 4, this section details the underlying algorithm and implementation used in our experiments.

**Step-by-step Procedure.** Given an annotated image $(x, y)$ with bounding box $b = (x_{\min}, y_{\min}, x_{\max}, y_{\max})$, we define the center $p \in \mathbb{R}^2$ of the box as the base fixation point. Glimpses are then constructed as follows:

- **Step 1: Sampling fixation candidates.** Around $p$, we sample up to $k_{\text{cand}}$ candidate centers $p_i$ using a uniform offset in both spatial directions. The maximum offset is set to a fraction $\alpha$ of the bounding box width/height, i.e.,

$$\Delta x, \Delta y \sim \mathcal{U}(-\alpha w, \alpha w), \quad \text{where } w = x_{\max} - x_{\min}.$$

  These jittered candidates simulate parafoveal fixations while remaining near the object center. Concretely, this defines a square jitter window around the center of the bounding box, within which candidate fixation points $p_i = p + (\Delta x, \Delta y)$ are sampled.

- **Step 2: Valid fixation selection.** For each candidate $p_i$, we compute a crop region whose aspect ratio and scale are randomly jittered using multiplicative factors $\beta_x, \beta_y \sim \mathcal{U}(1 - \beta, 1 + \beta)$. We retain up to $k \leq 3$ valid crops whose regions lie entirely within image bounds. This ensures all glimpses are valid, foreground-aligned views.

- **Step 3: Distortion-aware cropping.** For each selected $p_i$, the crop is resized to the model's input resolution. If the required resizing scale exceeds a threshold computed via an inverse crop ratio $\eta = 1/(1 - \texttt{max\_crop\_ratio})$, we first expand the crop window proportionally around its center (without crossing image bounds). This reduces geometric distortion when handling small or thin boxes.

- **Step 4: Aggregation.** Each image yields $k$ foveated crops $\{\text{Fov}_i(x, p_i)\}_{i=1}^{k}$. These are treated as label-consistent training samples and either randomly subsampled ($k = 1$) or stacked into a correlated mini-batch. Glimpses from the same image are never shuffled across batches, preserving the coherence of multi-view supervision.

**Implementation Notes.** The algorithm is implemented as a configurable pipeline which exposes key parameters:

- `Offset_fraction` $= 0.2$: sets $\alpha$, the maximum offset for sampling.

- `Scale_jitter` $= 0.1$: sets $\beta$, the jitter range for scale and aspect ratio.

- `Max_crop_ratio` $= 0.2$: defines threshold $\eta$ to trigger crop expansion. The max crop ratio is a threshold parameter that controls how much a crop is allowed to be resized relative to the original bounding box before geometric distortion is considered too high.

- `Area_threshold` $= 0.2$: used to activate distortion-aware expansion for small objects.

- `Multi_crop` flag: if `True`, all $k$ glimpses are returned together; if `False`, one random crop is sampled per epoch.

- `Augmentation mode` entails {"conservative", "medium", "aggressive"}: scales the above hyperparameters accordingly.

This design ensures that glimpses maintain semantic alignment while providing spatial diversity around the object. The same framework supports single-glimpse ($k = 1$) and multi-glimpse ($k > 1$) supervision via a unified pipeline.

Table 5: FocL dataset and cropping hyperparameters.

| Parameter | Value |
|---|---|
| Offset fraction ($\alpha$) | 0.2 |
| Scale/aspect jitter ($\beta$) | 0.1 |
| Max crop ratio | 0.2 |
| Area threshold (for distortion-aware fallback) | 0.2 |
| Number of glimpses $k$ | 1 or 3 |
| Multi-crop batching | Enabled for $k > 1$ |
| Batch size | 128 (k=1), 64 (k=3) |
| Input resolution | $224 \times 224$ |
| Augmentation | **Medium** |

### A.3 Training Details and Reproducibility

**Dataset Preparation.** Following the setup in Meehan et al. (2023), we sample and curate our annotated dataset from ImageNet using the official codebase available at `https://github.com/facebookresearch/DejaVu`. All dataset checks, bounding box extraction, and curation pipelines were built on top of this repository. We adapt their utilities to generate the subset used for FocL, ensuring consistency in annotation quality and reproducibility of bounding box metadata.

We evaluate FocL across multiple ImageNet subsets with bounding box annotations. Our experiments use the following curated partitions:

- **Full-scale split.** We use the complete curated bounding-box subset of ImageNet-1K (2012), totaling 482,187 images. We apply a 94/6 train–validation split, yielding 453,254 training images and 28,933 validation images. This **full-scale** setup is used for all generalization experiments presented in Section 4.2.

  For comparison, Section 4.3 also includes results from a standard ResNet-50 trained on the full ImageNet-1K classification dataset (approximately 1.3M images) using an 80/20 train–validation split, producing roughly 1.03M training examples. This model is included solely to show that increasing the volume of training data does *not* improve generalization on foveated (oracle-crop) inputs, as the standard model continues to rely heavily on non-foreground contextual cues.

- **Controlled low-data splits.** We also construct two disjoint 100K ImageNet subsets, referred to as **Partition A** and **Partition B**, each split into 85K training and 15K validation images. Partition A is used for all controlled analyses in reduced-data settings:

  - The 85K training split of Partition A is used to evaluate adversarial robustness (PGD perturbation distance), gradient norms, and memorization via cumulative sample loss (CSL), as discussed in Section 4.1.
  - The 15K validation split of Partition B is used for validation-time adversarial robustness evaluation, enabling a clean separation between training and evaluation subsets.

Each image is preprocessed to extract either one or up to three foveated crops using the method described in Section 3.1. All crops are resized to $224 \times 224$ resolution. Inputs are normalized using the standard ImageNet mean and standard deviation.

**Model Architecture**   We use a standard ResNet-50  (He et al., 2015) architecture across all experiments, with no architectural differences between Standard and FocL models.

**Training Configuration.**   All models are trained for 90 epochs using SGD with momentum 0.9 and weight decay $1 \times 10^{-4}$. The initial learning rate is set to 0.1 and decayed by a factor of 0.1 every 30 epochs. We use a batch size of 64 *per worker*, which is flattened across multiple glimpses during multi-crop training (e.g., $k = 3$ glimpses per image). We use a batch size of 128 for $k = 1$ glimpse per image. Each training sample is augmented using standard ImageNet transforms: random resized crop, horizontal flip, and color jitter. All experiments are tracked using Weights & Biases.

**Optimization and Learning Rate Schedule.**   We use the standard cross-entropy loss as the training objective. Optimization is performed using stochastic gradient descent (SGD) with momentum set to 0.9 and weight decay of $1 \times 10^{-4}$. The initial learning rate is 0.1, decayed by a factor of 0.1 every 30 epochs using a StepLR scheduler. All models are trained with mixed precision using PyTorch's `GradScaler` for improved stability and efficiency.

**Reproducibility and Statistical Significance**   We ensure statistical rigor by repeating key experiments across multiple random seeds and reporting mean and standard deviation where applicable:

- **Generalization experiments (Section 4.2):** All models evaluated using both oracle bounding box inference, FALcon and SAM inference are trained across 3 random seeds. We report the mean Top-1 accuracy along with standard deviation in the main paper.

- **Data-efficient learning (Section 4.3):** To assess consistency in low-data settings, we train models on Partition A (100K subset) across 5 different random seeds. The required table with standard deviation is provided in the Appendix.

- **CSL and adversarial robustness (Section 4.1):** Cumulative sample loss is computed by logging per-sample training loss across all 90 epochs on the 100K subset. We also evaluate PGD-based adversarial distance across 5 different $\ell_2$ budgets ($\epsilon$) on the same partition.

**Compute and Environment.**   All models are trained on NVIDIA A40 GPUs with 48GB memory per device. We follow the same training hyperparameters and optimization settings for both Standard and FocL models. The full training pipeline, configuration scripts, and an environment file are included in the code submission.

*An environment file named requirements.txt is included in the supplementary materials to ensure full reproducibility.*

Table 6: **Ablation under oracle bounding box inference on 2,000 ImageNet-V1 validation samples.** Top-1 and Top-5 accuracy (%) for three variants of the FocL training pipeline: the base foveated crop, a distortion-aware single-crop variant (SC), and the full multi-glimpse model (MC).

| Metric | FocL Base | FocL SC | FocL MC |
|---|---|---|---|
| Top-1 (%) | 69.05 | 71.05 | **72.23** |
| Top-5 (%) | 89.50 | 89.80 | **90.82** |

Table 7: **Ablation under SAM-2 proposals on ImageNet-V1 (in-distribution) and ImageNet-V2 (out-of-distribution).** "Any" Top-1 accuracy (%) is reported, where an image is counted correct if *any* SAM-generated crop is classified correctly.

| Dataset | FocL Base | FocL SC | FocL MC |
|---|---|---|---|
| ImageNet-V1 | 70.30 | 75.05 | **75.82** |
| ImageNet-V2 | 61.50 | 64.80 | **65.80** |

## A.4 Generalization Results

**A. Ablation: Role of Glimpse Diversity and Distortion-Aware Cropping.** We evaluate the contribution of each component of FocL on the same 2,000 ImageNet-V1 samples used in the main paper. All ablation experiments are reported using a single seed for clarity. The **FocL Base** model uses a single resized bounding-box crop with no jitter, distortion-aware expansion, or parafoveal variation. The **FocL Single-Crop (SC)** model corresponds to the full FocL algorithm with $k = 1$, introducing scale jitter, mild positional shifts, and distortion-aware cropping. The **FocL Multi-Crop (MC)** model uses $k = 3$ jittered glimpses, adding structured multi-view diversity.

As shown in Tables 6 and 7, performance improves consistently from Base to SC to MC across all evaluation settings. Under *oracle* bounding-box evaluation, SC already improves upon the Base model by **+2.00 pp** in Top-1 accuracy (69.05% → 71.05%), and MC yields a total gain of **+3.18 pp** (72.23%). These gains occur even under idealized cropping, where the input contains minimal background contamination.

Under the more realistic SAM-2 proposal setting, where crops may be noisy or incomplete, the improvements become substantially larger. On ImageNet-V1, SC improves Base by **+4.75 pp** (70.30% → 75.05%), and MC yields a total gain of **+5.52 pp**. On ImageNet-V2, an out-of-distribution benchmark, SC improves Base by **+3.30 pp**, and MC by **+4.30 pp**. These trends demonstrate that both *distortion-aware cropping* (SC) and *structured glimpse diversity* (MC) play complementary roles: the former increases robustness to imperfect box geometry, while the latter improves stability under viewpoint variation and proposal noise.

**B. Diagnostic analysis of background dependence.** To better understand the failure modes of the Standard ResNet-50, we visualize a set of images that are correctly classified when evaluated on the full ImageNet input, but misclassified when the model is evaluated only on the ground-truth object crop. Figure 9 shows representative examples across diverse categories. In many cases, the full image prediction is driven by strong scene-level correlations—such as characteristic backgrounds (e.g., water, vegetation), co-occurring objects (e.g., humans, tools), or global geometry and scale, which are removed when only the object bounding box is provided. The resulting tight crops often preserve only ambiguous local texture or partial shape, leading to confident but incorrect predictions. These qualitatively illustrate that the standard model does not acquire robust object-centric features and instead leans heavily on background cues.

**C. Additional COCO cross-domain results.** To supplement the COCO experiments in the main paper, we provide additional qualitative predictions from the Classifier + SAM-2 system in Figure 10. These examples span a range of proposal budgets and scene complexities, including occlusion and dense clutter. Consistent with the main findings, the FocL classifier selects the correct SAM proposal substantially more reliably than the Standard model, which frequently relies on background correlations when transferred to

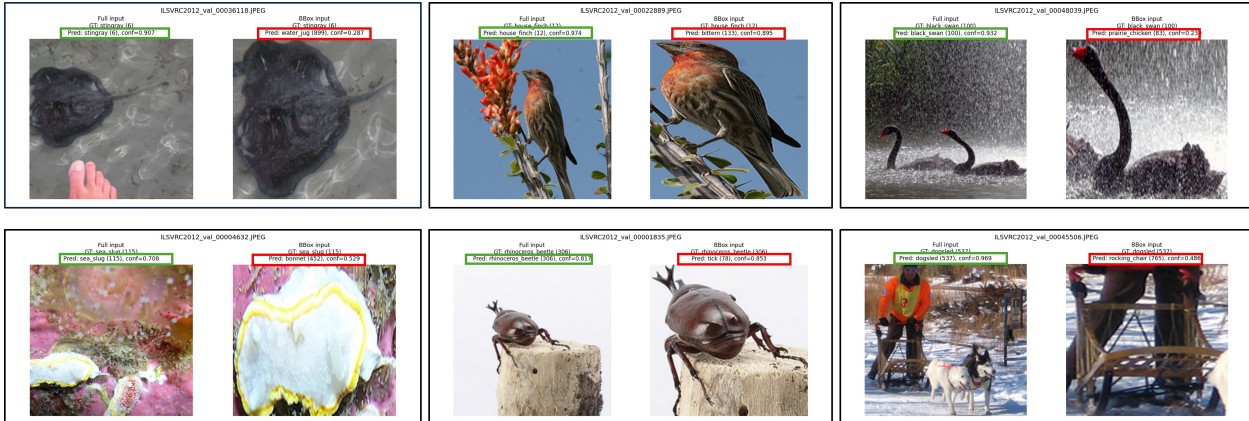

Figure 9: **Representative failure cases showing the dependence of a Standard ResNet-50 on background context.** Each pair shows the full image (left) and the corresponding ground-truth bounding-box crop (right). While the model predicts the full image correctly (green), removing background features often eliminates global scene cues such as habitat, co-occurring objects, scale, or pose, causing confident misclassifications on the cropped object (red). These examples indicate that standard ImageNet training does not yield robust object-centric representations.

Table 8: **Core SAM-2 hyperparameters for ImageNet proposal generation (20 proposals/image).**

| Parameter | Value | Purpose |
|---|---|---|
| SAM version | SAM-2.1 (Hiera-L) | Mask proposal backbone |
| pred_iou_thresh | 0.5 | Retain lower-confidence masks for diversity |
| stability_score_thresh | 0.5 | Relaxed region filtering |
| min_mask_region_area | 25 px | Allow small objects |
| NMS IoU threshold | 0.9 | Keep diverse proposals |
| Max proposals | 20 | Top proposals used for classification |
| Fallback | Full image | Used if SAM yields no valid masks |

COCO. These qualitative results reinforce FocL's improved cross-domain generalization and its ability to operate robustly under noisy object proposals.

**D. SAM-2 hyperparameters for ImageNet proposal generation**  For the ImageNet and ImageNet-V2 cross-domain experiments using the Classifier + SAM system, we generate object proposals with SAM-2.1 (Hiera-Large) in automatic mask generation mode. To encourage diverse candidate regions suitable for object-centric classification, we use relaxed confidence thresholds and retain the top 20 proposals per image. Masks with extremely small foreground support (area ratio $< 10^{-4}$) or bounding boxes smaller than $5 \times 5$ pixels are filtered out. A high NMS IoU threshold (0.9) is applied to remove only near-duplicate proposals, ensuring that viewpoint-, scale-, and boundary-level variations are preserved. If no valid proposal remains after filtering, we fall back to using the full image as a single proposal.

The resulting proposal distribution matches the operating regime used in the main paper, where the Standard and FocL classifiers are evaluated on ImageNet-V1 and ImageNet-V2 using 20 SAM-2 proposals per image. Table 8 summarizes the key hyperparameters.

**E. Classifier + SAM evaluation (Any prediction)**  For the Classifier + SAM experiments on ImageNet-V1 and ImageNet-V2, we evaluate the models using an "Any" prediction metric built on top of the SAM-2 proposals described above. For each image, SAM-2 provides up to 20 candidate bounding boxes, which we crop from the original resolution, resize to $224 \times 224$, normalize with ImageNet statistics, and classify in a single batch. The classifier produces a Top-1 prediction for every crop, and an image is

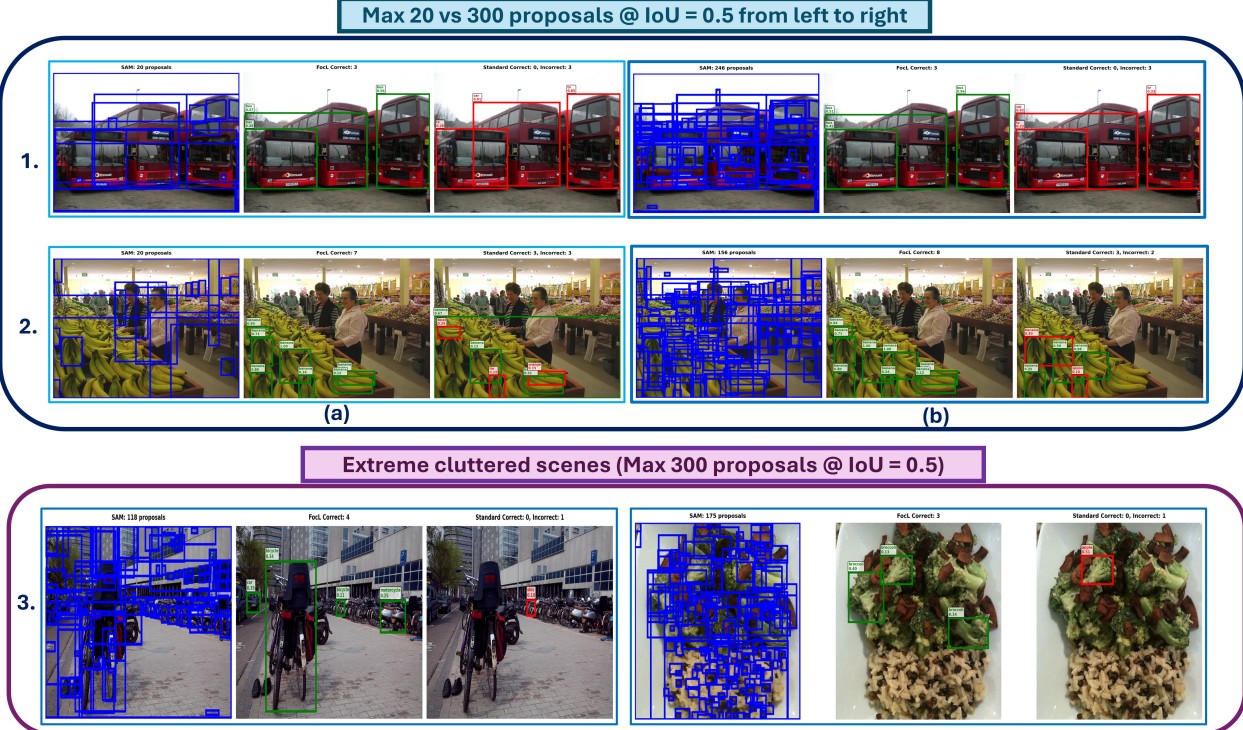

Figure 10: **Additional COCO cross-domain generalization examples using the Classifier + SAM system.** Following the analysis in the main paper, we visualize further cases where an ImageNet-trained classifier is paired with SAM-2 proposals and evaluated on COCO images. Across both low-proposal (20) and high-proposal (300) settings, FocL reliably selects the correct object-centric proposal even in the presence of clutter or heavy proposal noise, whereas the Standard classifier often fires on contextual or background-driven regions. These examples complement the quantitative cross-domain results reported in the main text and further highlight the robustness of FocL to domain shift and proposal imperfections.

Table 9: **SAM-2 settings for COCO proposal generation.** Both regimes use the same mask-generation thresholds; only the number of retained proposals differs. Multiscale inference is applied for the 300-proposal setting to increase spatial and scale diversity.

| Parameter | COCO-20 | COCO-300 |
|---|---|---|
| SAM version | SAM-2.1 (Hiera-L) | SAM-2.1 (Hiera-L) |
| Pred. IoU threshold | 0.70 | 0.70 |
| Stability threshold | 0.80 | 0.80 |
| Min. region area | 100 px | 100 px |
| NMS IoU threshold | 0.65 | 0.65 |
| Multiscale inference | No | Yes (0.85, 1.0, 1.15) |
| Max proposals per image | 20 | 300 |

counted as correct under the Any metric if any crop-level Top-1 prediction matches the ground-truth ImageNet label. Because this metric assumes ideal proposal selection, it serves as an empirical high performance ceiling on the performance of the Classifier + SAM system. Images for which SAM-2 yields zero valid proposals are kept in the denominator and counted as incorrect. This evaluation is applied identically to the Standard and FocL classifiers, ensuring a fair comparison, and the resulting Any accuracies correspond to the values reported in Table 7 and in the main paper.

**F. SAM-2 hyperparameters for COCO proposal generation**   For COCO cross-domain experiments, we generate proposals using SAM-2.1 (Hiera-L) in automatic mask generation mode. We consider two operating points that correspond to the settings used in the main paper: a **20-proposal regime** for lightweight evaluation and a **300-proposal regime** which emphasizes coverage in cluttered scenes. Both settings use the same SAM-2 configuration, differing only in the maximum number of retained proposals.

To ensure comparability with the ImageNet experiments, we use consistent mask-generation thresholds and filter out only extremely small connected components. For each image, all SAM-2 masks are converted to bounding boxes, merged using IoU-based non-maximum suppression, and then ranked by SAM's predicted IoU score. We keep the top $N \in \{20, 300\}$ proposals. When multiscale inference is enabled (for the 300-proposal setting), proposals from three resized versions of the image $(0.85\times, 1.0\times, 1.15\times)$ are merged before final selection. No ground-truth information is used at any stage.

**G. Cross-domain generalization on COCO with the Classifier + SAM system.**   To assess the cross-domain behavior of the Classifier + SAM system, we evaluate ImageNet-trained Standard and FocL models on COCO using pre-computed SAM-2 proposals (20–50 per image). Following the mapping protocol described in the main paper, we restrict evaluation to COCO categories that have either a direct or super-category correspondence with ImageNet. For each proposal, we crop the region, apply standard ImageNet preprocessing, and assign a COCO category by taking the maximum softmax score over all ImageNet classes mapped to that category. We then apply a fixed confidence threshold (0.1) and per-category non-maximum suppression (IoU 0.5). Detection performance is measured using the standard 11-point interpolated average precision at IoU thresholds 0.3 and 0.5. No COCO labels or bounding boxes are used for training, tuning, or proposal filtering. This evaluation isolates how well each ImageNet classifier transfers to COCO under identical SAM proposals, providing a clean measure of cross-domain generalization.

**H. Note on NMS thresholds.**   For proposal generation, we apply a relaxed NMS IoU threshold of 0.65 to preserve spatial and scale diversity among SAM-2 candidates. During COCO evaluation, however, detections are post-processed using the standard per-category NMS with an IoU threshold of 0.5, consistent with conventional detection protocols.

## A.5   Adversarial Robustness

**Setup**   We evaluate adversarial resistance by computing the minimum $\ell_2$ perturbation required to flip model predictions using PGD attacks (Madry et al., 2018). All experiments are conducted on a balanced ImageNet subset with 100 samples per class and an 85/15 train-validation split. We use a PGD-$\ell_2$ attack with 10 steps, random initialization, and random restarts enabled. The step size is set to $\alpha = \epsilon/10$, and we sweep the perturbation budget $\epsilon \in \{0.0, 0.25, 0.5, 0.75, 1.0\}$.

To construct a clean and balanced evaluation protocol, we select 15,000 correctly predicted samples from the training set (Partition A) and 15,000 correctly predicted samples Partition B. This forms the validation set results for Partition A (unseen). This ensures that the evaluation is based on semantically aligned, clean samples and keeps the number of inputs consistent across training and validation settings. We compute both the robustness curves and the mean adversarial distance on these subsets, allowing for a fair comparison between FocL and standard models.

**Mean Adversarial Distance**   To quantify robustness, we compute the average adversarial distance:

$$\bar{d} = \frac{1}{N} \sum_{i=1}^{N} \|\delta_i\|_2 \quad \text{where} \quad f(x_i + \delta_i) \neq y_i$$

Here, $\delta_i$ denotes the smallest perturbation (in $\ell_2$ norm) found via PGD that causes a misclassification. We find that the standard model has $\bar{d} = 0.3806$, while FocL achieves $\bar{d} = 0.6169$—a 62% increase. This gap reflects a substantial improvement in robustness. Higher adversarial distance implies that more energy is required to change the model's decision, suggesting a more stable and semantically aligned representation. These results support the argument that standard models overfit to incidental background cues, while FocL focuses learning on foreground-relevant features that are inherently harder to perturb.

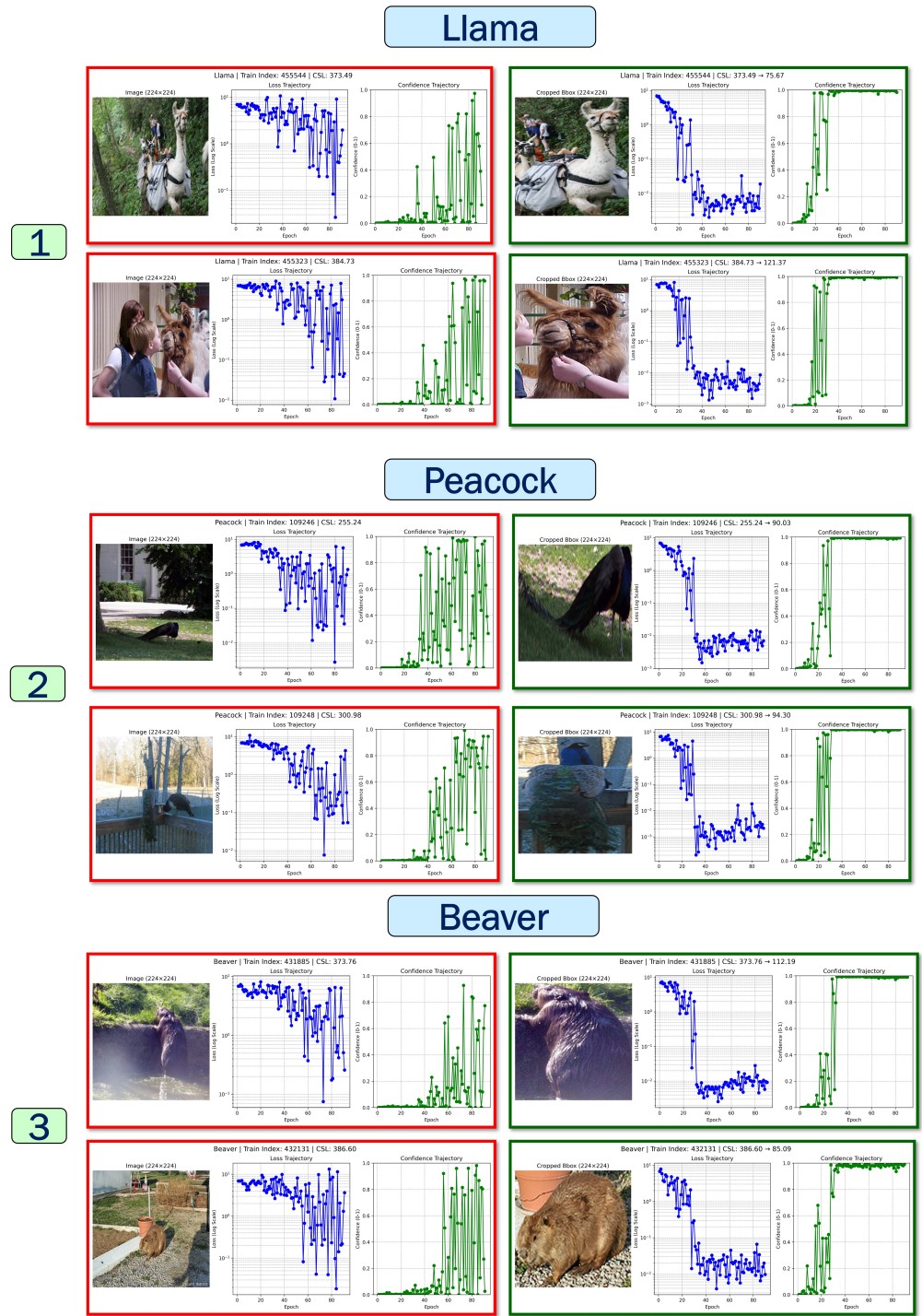

Figure 11: Sample visualization of CSL dynamics across three classes—**Llama**, **Peacock**, and **Beaver**. Each row compares full-image (left, red box) vs. FocL-based cropped inputs (right, green box). Across classes, FocL leads to faster convergence (loss trajectory), more confident predictions (confidence trajectory), and substantially lower cumulative sample loss (CSL). These patterns are consistent with aggregate statistics shown in CSL distributions, train-validation loss curves, and gradient norm plots.

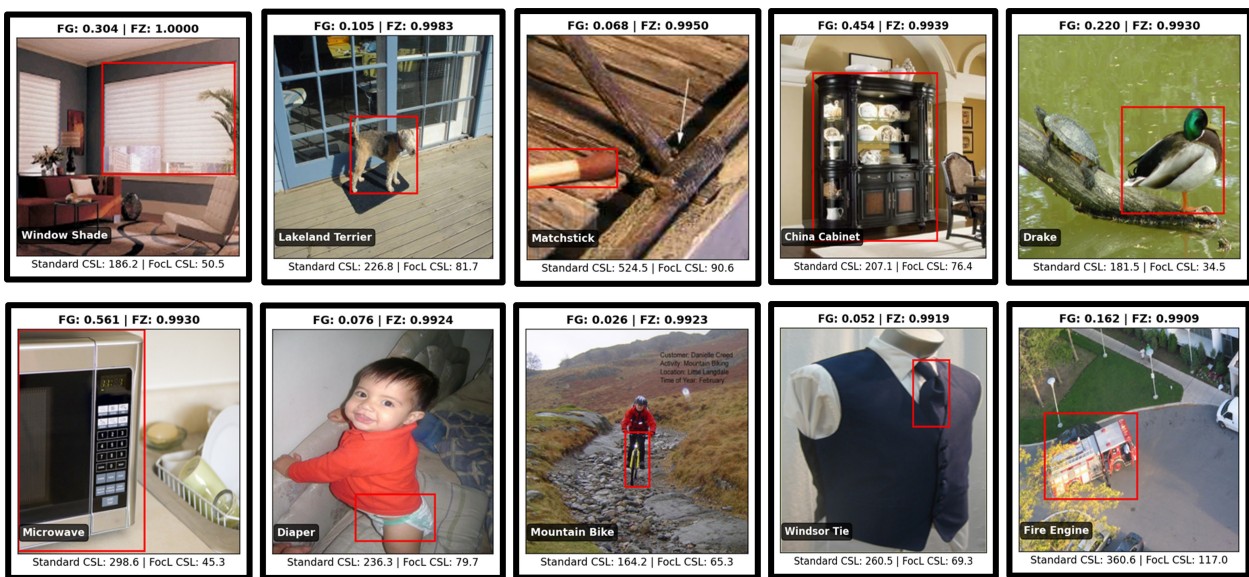

Figure 12: **Additional examples from the Feldman–Zhang (FZ) memorization cohort.** Shown are samples identified as highly memorized according to the FZ score, together with their bounding-box annotations and CSL values for the Standard and FocL models. Some exhibit severe background clutter or ambiguous multi-object scenes. We report FG only as a descriptive attribute (box-area / image-area), not as evidence for memorization. As in the main paper, FocL consistently reduces CSL on these difficult samples, indicating substantially lower memorization.

### A.6 Cumulative Sample Loss (CSL) as a proxy for learning difficulty

**Setup** We evaluate cumulative sample loss (CSL) as a proxy for sample difficulty and memorization. The setup follows the same 85K/15K train-validation split used in the robustness analysis. CSL quantifies how difficult a sample is to learn by accumulating its training loss over epochs. Formally, for a training sample $z = (x, y)$, the cumulative sample loss over $T$ epochs is defined as:

$$\text{CSL}(z) = \sum_{t=1}^{T} \mathcal{L}(f_{\theta_t}, z)$$

where $\mathcal{L}$ denotes the cross-entropy loss, $f_{\theta_t}$ is the model at epoch $t$, and $z$ is the training sample. For the FocL model, the sample is represented as $\text{Fov}(x, y)$, denoting a foveated crop centered on the object. High CSL values correspond to samples that remain difficult across multiple epochs and are more likely to be memorized rather than learned robustly. For fair comparison, we evaluate CSL for the FocL single-crop variant to match the standard model's single-view training.

In Figure 11, we provide a per-sample analysis demonstrating how FocL facilitates easier learning. For each class (Llama, Peacock, Beaver), the green-boxed examples (FocL) consistently show lower cumulative sample loss (CSL) compared to their red-boxed full-image counterparts. This shift in CSL values explains the leftward shift in the aggregate CSL distribution observed in the main paper, supporting our claim that FocL improves learning stability and efficiency.

**Additional FZ memorization examples.** To complement the analysis in Section 4.1, we provide further qualitative examples from the Feldman–Zhang memorization cohort in Figure 12. These examples often include small object extent and substantial surrounding context; they are provided for illustration of the FZ-selected cohort. A notable case is the fifth image in the first row, where a *drake* and a *box turtle* co-occur within the same scene despite ImageNet assigning only a single label. This is a canonical instance of the label ambiguity highlighted in our first figure in the main text, where the Standard model overfits to correlated background structure. Across all shown cases, FocL markedly reduces CSL relative to the Standard model,

including challenging cases with cluttered or ambiguous scenes. These additional examples reinforce the central implication of our FZ analysis: FocL systematically reduces CSL on challenging, cluttered, and label-ambiguous samples in the FZ-selected cohort, shifting memorized instances closer to the mode of the difficulty distribution.

**Clarifying the p-value in Sec. 4.1 (memorization reduction).** We define a sample as "easier to learn" if its cumulative sample loss is lower under FocL than under Standard for the same image (paired by sample index on the same training split), i.e., $\mathrm{CSL}_{\mathrm{FocL}} < \mathrm{CSL}_{\mathrm{Std}}$. In the top-1% Feldman–Zhang memorization cohort overlapping our Partition-A training set ($n = 820$), 819/820 samples improve (99.88%). We assess this with a **one-sided exact binomial (sign) test** under the null that improvement is equally likely as degradation ($H_0 : p = 0.5$ for $\mathrm{CSL}_{\mathrm{FocL}} < \mathrm{CSL}_{\mathrm{Std}}$), which yields $p = 1.17 \times 10^{-244}$. As a complementary magnitude-based summary, we also report a **paired t-test** on CSL differences (mean CSL drops from 204.37 to 72.22; $p = 1.09 \times 10^{-276}$).

**FG ratio diagnostic.** FG denotes the *foreground area fraction* (ground-truth bounding-box area divided by full image area). We report FG only as a descriptive metric to contextualize the memorized cohort; it is not used as a causal proxy for memorization. On the 85K training split, the baseline mean FG is 0.470, while the memorized cohort mean is lower (0.457), this is a small difference. Furthermore, the fraction of samples with FG < 0.5 differs by only 1.33 absolute percentage points between the memorized cohort and the full 85K split. These statistics demonstrate that FG is a weak separator and does not explain cohort membership. Consequently, our memorization conclusions rely strictly on the identified cohort and the significant CSL reductions. Qualitative visualizations are included to illustrate the contextual complexity of memorized samples, not as evidence that FG drives memorization.

## A.7 Gradient Norm Analysis

**Setup** To probe the optimization dynamics of FocL, we analyze the magnitude of gradients during training. Specifically, we compute the $\ell_2$ norm of gradients with respect to all model weights on the training set of Partition A (85K samples from the 100K ImageNet subset). Gradient norms are logged throughout training for both the standard model and the FocL single-crop variant. This analysis provides insight into training stability and the ease of optimization under different input regimes.

FocL exhibits consistently smaller gradient magnitudes compared to standard training, suggesting a smoother optimization landscape. The standard model records a mean gradient norm of $3.81 \times 10^4$ with a standard deviation of $2.26 \times 10^4$, while FocL reports a lower mean of $2.07 \times 10^4$ and a standard deviation of $1.28 \times 10^4$. When normalized by the total number of ResNet-50 parameters ($\sim 2.56 \times 10^7$), the per-parameter gradient norm drops from $1.49 \times 10^{-3}$ (standard) to $8.08 \times 10^{-4}$ (FocL)—a relative reduction of approximately 45.8%. This substantial drop suggests that FocL's object-centric inputs result in less gradient noise and more stable optimization, aligning with our findings on faster convergence and lower memorization.

## A.8 Data efficiency

**Setup for Low-Data Regime** To evaluate data efficiency, we train all models on Partition A of the 100K balanced ImageNet subset. The standard baseline uses 85 training samples per class, while low-data models are trained with 50 samples per class. These 50-per-class subsets are derived from five random data partitions of Partition A (i.e., five distinct data seeds). All models are trained for 90 epochs using SGD with momentum and a step learning rate scheduler (decay at epochs 30 and 60), with otherwise identical hyperparameters.

Evaluation is performed on a fixed 50K test set from Partition B. The standard model is evaluated on full-resolution images, while FocL models are evaluated using bounding box–aligned crops. As shown in Section 4.1.1 of the main paper, full-image models underperform when evaluated on oracle bounding boxes. Therefore, we report results using their respective optimal evaluation inputs. For FocL, we use the single-crop variant, consistent with the setup in Section 4.3.

Table 10: Evaluation in the low-data regime using the 50K test set from Partition B. With 41.18% fewer training samples (50 vs. 85 per class), the FocL Single Crop model achieves comparable or better performance than the standard model trained on the full set. All results report mean ± standard deviation across 5 random data partitions.

| Tested on | Dataset Size (K) | Top-1 | Top-5 |
|---|---|---|---|
| Full Image (Standard) | 85 | 44.30 | 68.56 |
| Full Image (Standard) | 50 | $26.91 \pm 1.20$ | $49.36 \pm 1.17$ |
| Bounding Box (FocL SC) | 50 | $45.04 \pm 0.93$ | $70.30 \pm 0.85$ |

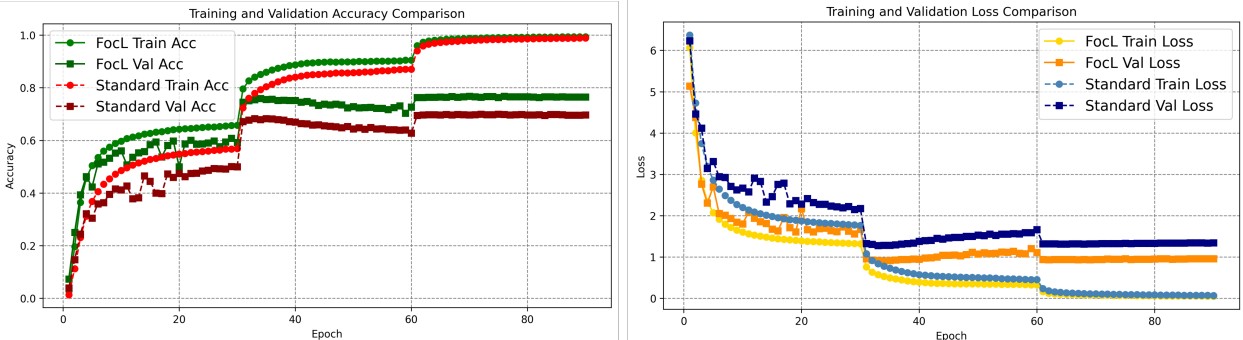

Figure 13: Training and validation accuracy and loss curves for FocL and standard models over 90 epochs. FocL converges faster and achieves lower training and validation loss throughout.

**Analysis.** FocL demonstrates superior data efficiency in the low-data regime as well. With only 50 training samples per class (41.18% fewer than the standard baseline), it achieves a **Top-1 accuracy of 45.02%** and **Top-5 of 70.30%**, outperforming the standard model trained on 85 samples/class (Top-1: 44.30%, Top-5: 68.56%). This margin holds consistently across 5 random data partitions. **These results are statistically consistent across data splits, highlighting FocL's robust ability to leverage object-centric signals even in lower data regime.**

### A.9 Training Dynamics and Convergence.

Learning curves on the 482K ImageNet subset (Figure 13) highlight the impact of foveated inputs on optimization behavior. Across the entire training trajectory, FocL converges faster, reaches lower training and validation losses, and achieves higher accuracy than the Standard model at matched epochs. This indicates that the foveated representation prior reduces the effective complexity of the learning problem, enabling the optimizer to make more stable progress with fewer samples. The gap emerges early in training, within the first few epochs, and remains consistent throughout, suggesting that FocL benefits from a smoother loss landscape and more informative gradients. Overall, these trends support the view that object-centric inputs lead to improved sample efficiency, more predictable optimization dynamics, and better generalization from the same amount of data.

### A.10 Comparison with Feature Pooling (Two-Stage Detection Baseline)

To validate the motivation behind the FocL "foreground-only" strategy, we compare it against a "localize-then-classify" baseline modeled after the second stage of modern two-stage object detectors (e.g., Faster R-CNN). This baseline employs feature pooling rather than raw image cropping to isolate object information.

1. **Dataset:** We utilize the curated ImageNet subset (482K images) consistent with the main paper. This ensures that the model is exposed to the same object-centric instances and ground-truth bounding box distributions as the FocL and Standard models.

2. **Baseline Method:** We implement a ResNet-50 backbone modified for feature pooling. The architecture processes the full-resolution image up to the third residual block (Layer 3), resulting in a feature map with a spatial stride of 16. We then apply an RoI Align layer (`torchvision.ops.roi_align`) with a $7 \times 7$ output spatial size and a spatial scale of 1/16. This allows the model to extract region-specific features directly from the high-dimensional feature map based on ground-truth coordinates. The pooled features are passed through a global average pooling layer and a final linear classifier.

3. **Evaluation:** same hyper-parameters as the standard and FocL models.

This comparison isolates the effect of the input representation (hard foveated crops vs. pooled feature maps). While feature pooling is the standard for detection, our results demonstrate that providing the model with high-resolution "hard" crops (FocL) leads to superior classification accuracy and reduced memorization of background noise compared to pooling from a lower-resolution backbone feature map.

### A.11  Evaluation of Background Removal without Resolution Enhancement

To evaluate whether the performance gains in FocL stem solely from the removal of background clutter, we compare it against a "HardMask" baseline. This model processes images where the background is strictly neutralized while the object remains at its original image-space scale, thereby isolating context removal from foveated resolution enhancement.

1. **Dataset:** We utilize the curated ImageNet subset (482K images) consistent with the main paper. This ensures the model is trained on identical instances and ground-truth bounding box distributions as the FocL and Standard models.

2. **Baseline Method:** We implement a "Mean-Pad" masking strategy within a custom dataset wrapper (`HardMaskDataset`). For each image, the ground-truth bounding box is parsed from ImageNet XML annotations. A composite image is then created at the original resolution: pixels inside the bounding box are preserved, while all pixels outside the box are replaced with a solid color representing the ImageNet dataset mean (RGB: 124, 116, 104). This masked image—containing the object at its original scale—is then passed through a standard augmentation pipeline (e.g., RandomResizedCrop) and a standard ResNet-50 backbone.

3. **Evaluation:** Same hyper-parameters as the standard and FocL models.

This baseline strictly isolates the impact of background removal. By maintaining the object at its original resolution within the full image frame (no zooming or foveation), this setup tests whether performance gains observed in FocL are merely a result of filtering background noise or if the foveated resolution enhancement provided by cropping and resizing is necessary for the observed improvements.

### A.12  Evaluation under Idealized Localization (Oracle GT)

The primary motivation for this experiment is to assess the inherent classification capability of each training strategy when provided with perfect localization information. By utilizing oracle ground-truth bounding boxes, we isolate the performance of the ventral classifier from any potential errors introduced by a dorsal proposal generator.

**Inference Methodologies:**

- **Standard:** Inference is performed on the full-resolution image using standard center-cropping, representing a traditional classification pipeline.

- **RoI Align:** The model processes the full $224 \times 224$ image, and features are extracted from the ground-truth region via RoI Align pooling on the Layer 3 feature map, exactly replicating its training mechanism.

Table 11: Classification performance comparison on the 2K ImageNet-V1 subset using Oracle ground-truth (GT) boxes. All models are evaluated in their optimal inference domain to assess the inherent capability of each training strategy.

| Method | Inference Strategy | Top-1 (%) | Top-5 (%) |
|---|---|---|---|
| Standard | Full image (standard classifier inference) | 60.90 | 83.05 |
| RoI Align | Full image + GT box (RoIAlign pooling on GT boxes) | 63.55 | 84.65 |
| HardMask | GT box + mean-pad outside (HardMask inference) | 62.95 | 85.95 |
| FocL (SC) | GT box cropping inference | **71.05** | **89.80** |
| FocL (MC) | GT box cropping inference | **72.23** | **90.82** |

- **HardMask:** Ground-truth regions are preserved while the background is masked with the dataset mean color at the original resolution, followed by a standard validation transform, matching the HardMask training protocol.

- **FocL (SC/MC):** The ground-truth bounding box is used to extract a foveated crop which is resized to the input resolution for classification.

**Results**   The empirical results in Table 11 demonstrate significant performance disparities across the evaluated strategies. While the RoI Align and HardMask baselines improve upon the Standard model by 2.65 and 2.05 percentage points in Top-1 accuracy, respectively, they remain substantially inferior to the FocL models. Specifically, FocL (SC) achieves 71.05% Top-1 accuracy, outperforming the RoI Align baseline by 7.5 percentage points and the Standard model by 10.15 percentage points. FocL (MC) further extends this lead to a 72.23% Top-1 accuracy.

**Explanations**   The modest gains in HardMask and RoIAlign are expected, since both leverage localization to emphasize the foreground and reduce reliance on spurious background correlations. However, the RoIAlign baseline pools from a backbone feature map computed on the full image; because these features have large receptive fields (and are influenced by global context through convolution/padding/normalization), the resulting RoI features are not strictly object-only and can still carry contextual signals. In contrast, HardMask removes background in pixel space but does not increase the object's effective resolution, since the object remains at its original scale within the resized frame. FocL addresses both issues by explicitly foveating in image space (crop + resize), providing higher-resolution object evidence to the classifier, which accounts for the remaining performance gap beyond context removal alone.

## A.13   Evaluation under Realistic Localization (SAM-2 Proposals)

The primary motivation for this experiment is to evaluate the transferability and robustness of the ventral classifiers within a realistic active vision pipeline. By pairing the classifiers with the Segment Anything Model (SAM-2) as a dorsal proposal generator, we test their capability to handle real-world proposal noise and natural distribution shifts on ImageNet-V1 and ImageNet-V2. This evaluation utilizes the "Any Accuracy" metric, which serves as an empirical performance ceiling for the overall system by considering a sample correct if the ground-truth label is predicted for any of the top 20 proposals.

Following the identical evaluation setup and hyperparameters outlined in Appendix A.4.D and E, all models are evaluated using the same set of SAM-2 generated proposals to ensure a fair comparison. For the Standard and FocL models, the inference strategy follows the localized object crops pipeline. To provide a rigorous baseline comparison, the RoI Align and HardMask models are evaluated using both their native training-matched strategies and the localized object crops strategy.

**Inference Methodologies:**

- **Standard:** The inference strategy follows the localized object crops pipeline where each SAM proposal region is cropped from the original image and resized to the input resolution.

Table 12: Classification performance under the SAM-2 proposal regime using the "Any Accuracy" metric. This evaluates the robustness of the classifiers when paired with an external dorsal localizer on ImageNet-V1 and ImageNet-V2.

| Method | Inference Strategy | V1 Acc (Any) | V2 Acc (Any) |
|---|---|---|---|
| Standard ResNet | Localized Object Crops | 68.08% | 57.50% |
| HardMask | Proposal Masking (Hardmask) | 62.30% | 50.90% |
| HardMask | Localized Object Crops | 60.00% | 48.90% |
| RoI Align | RoI Pooling | 73.05% | 58.75% |
| RoI Align | Localized Object Crops | 59.45% | 49.25% |
| FocL (SC) | Localized Object Crops | **75.05%** | **64.80%** |
| FocL (MC) | Localized Object Crops | **75.82%** | **65.80%** |

- **RoI Align:** This baseline is evaluated using (1) RoI Pooling, which processes the full $224 \times 224$ image and extracts features for each SAM proposal via RoI Align pooling on the Layer 3 feature map to replicate its training mechanism, and (2) Localized Object Crops.

- **HardMask:** This baseline is evaluated using (1) Proposal Masking, which neutralizes the background of the full original image with the dataset mean color around each SAM box (followed by a standard validation transform without foveated zoom), and (2) Localized Object Crops.

- **FocL (SC/MC):** The inference strategy follows the localized object crops pipeline where each proposal region is cropped from the original image and resized to the input resolution.

**Results**  The empirical results in Table 12 demonstrate that FocL significantly outperforms all baselines under realistic proposal conditions, establishing its superiority in active vision tasks. While the RoI Align model achieves competitive performance using its native RoI Pooling strategy on ImageNet-V1 (73.05%), closing the gap to FocL (SC) to within 2%, its accuracy collapse on ImageNet-V2 is much more pronounced, with the gap extending to 6.05%. Similarly, when evaluated using Localized Object Crops, the performance of the RoI Align model drops to 59.45% on V1 and 49.25% on V2, indicating that its features are heavily coupled to the global context and fixed spatial resolution of the full-image feature map.

**Explanation**  FocL learns robust object-centric features that transfer superiorly across datasets because it is trained to identify objects in a scale-invariant, foveated input domain. By forcing the ventral stream to classify high-resolution, cropped instances, FocL prevents the model from relying on spurious background correlations or the specific spatial stride of a backbone feature map. Consequently, while feature-pooling baselines like RoI Align suffer when faced with the distribution shifts and proposal noise inherent in ImageNet-V2, FocL maintains its lead by leveraging high-acuity object evidence that remains consistent across different environments.

