# OpenReview forum: "From Clutter to Clarity: Visual Recognition through Foveated Object-Centric Learning (FocL)"
_TMLR — Accepted by TMLR_

### Review · Reviewer_vWJm · 2025-12-01

**Summary Of Contributions:**

The paper introduces a new method, called FocL, for learning a classifier from images with pre-annotated segmentations of the main object(s). The method consists of taking one or many crops of varying size centered near the central object's bounding box, and then using those crops directly as training samples in the same batch in the standard classification pipeline. The authors conduct experiments to show that this method improves over classifiers trained using the standard pipeline in several ways: training stability, reducing memorization, cleaner embedding spaces as measured by t-SNE projection, robustness to $\ell_{p}$ input perturbations, and sample efficiency. When segmentations are not available at inference-time, the authors show that FocL-trained classifiers can also work with a trained segmentation model such as SAM.

Main strengths:
- The FocL method is simple and intuitive: reducing background context leads to smoother training and fewer outliers, leading to most or all of the above mentioned benefits.
- Many of the above benefits (see below) are relatively significant on image data where a pre-annotated segmentation exists.

Main weakness:
- The method needs a pre-annotated bounding box around the relevant object(s), at least during training (as far as I can see, the use of a model to do segmentation during _training_ time has not been explored). This makes a comparison between FocL and a standard classification pipeline not very fair, because you are essentially comparing a model which maps image -> class to a model which maps (image, segmentation) -> class (or, more relevantly for comparing training stability, the training steps map (image, class) -> loss vs (image, class, segmentation) -> loss). There can be some simple baselines for a comparison in this sense, e.g., feeding in an embedding of the coordinates along with the whole image (if the authors were using a transformer this could be done using cross-attention or similar; the situation is slightly more delicate for the ResNet backbone that they do use).

**Audience:**

Yes

**Audience Explanation:**

The problem is to learn an image classifier, a standard task in machine learning and one in which many members of TMLR's audience may be interested. This paper argues that there is a substantial benefit to the image classifier when using segmentation maps during training and inference, at least using their method. If true this could potentially improve practical classification pipelines when there is segmentation data already available.

**Broader Impact Concerns:**

No concerns

**Claims And Evidence:**

Yes

**Claims Explanation:**

There are no theoretical claims. The empirical claims are mostly supported, with some caveats, which I will  go through in order of their presentation:
- FocL reduces memorization (4.1): The authors claim that FoCL makes 99.88% of the "hardest samples" (defined more precisely in the paper) easier to learn, then cites a $p$-value of $< 0.001$. What is this $p$ value? What is the underlying probability distribution, etc? Also, in the same paragraph, they cite that such hardest samples are hard because they have foreground to image area ratio (FG ratio) relatively large, saying it is equal to 0.457 in expectation. What is the baseline FG ratio?
- FocL improves generalization under foveated inputs (4.2): The authors claim that FoCL-trained models with a pretrained SAM to do the bounding box annotation perform better at in-domain image classification (training set ImageNet V1, testing set ImageNet V1) and out-of-domain image classification (training set ImageNet V1, testing set ImageNet V2 or COCO). The claim here is a bit tricky since SAM has surely been trained on many kinds of images including ImageNet V2 and COCO. However, using SAM with standard image classifiers still demonstrates the gap, so I think it should be OK. In my opinion, it would be better to also do some experiments with a possibly weaker segmentation model, where you know the training data exactly and can test on some truly OOD data.
- FocL enhances learning efficiency and stability (4.3): In instances where the authors vary the training hyperparameters (e.g. number of steps or learning rate schedule), they do not specify whether the hyperparameters are optimized (by hand or otherwise) for this setting. E.g. a suboptimally chosen learning rate schedule may explain the results of 4.3. If hyperparameters are only optimized for one case (like the main case) they can cause bad training dynamics in other cases. It might be worthwhile to clarify this, as I didn't see this in a read-through of the main body and appendix.

More broadly, a major focus of the paper is that FocL improves on the standard baseline classification _because_ it does not feed in background context into the model. This is not sufficiently proven, in my opinion. This is related to the weakness discussed above, and can be resolved in the same way: try some other simple baselines which incorporate segmentation information into the model and yet include the whole input image to the network. This would reveal whether the effect is due to the additional segmentation information or the loss of irrelevant information.

**Requested Changes:**

These changes derive from the above weakness and analysis of the main claims.
1) This has been suggested twice above: train some baselines which incorporate segmentation information to demonstrate that the improvements of FocL really do come from removing the irrelevant context. Unless I am missing something in my reading, this issue seems the most critical to me (and therefore for my recommendation for acceptance).

2) It would be good to clear up the numbered caveats pointed out in the analysis of the paper's claims. Namely, the first and third caveats, related to sections 4.1 and 4.3 respectively, can be done without any additional experiments and just clarified in the paper, if the results are as implied by the relevant sections of the paper. The second caveat, related to section 4.2, would be a nice-to-have-resolved but is not necessary for my acceptance due to the marginal value; it would also require extra effort to run such experiments with a new segmentation model (especially one trained just by the authors).

---

> ### Author Response · Authors · 2026-01-05
> **Rebuttal for Reviewer vWJm**
>
> We thank Reviewer vWJm for the detailed suggestions. The updated manuscript incorporates the requested changes in **blue**. In particular, we add intermediate baselines to isolate whether FocL’s gains arise merely from background/context removal: a **RoIAlign** feature-pooling baseline (Appendix A.10) and a control baseline **HardMask/MeanPad** (Appendix A.12) that neutralizes pixels outside the GT box without crop-and-zoom. We also provide supporting evaluations under **oracle GT-box** inputs (Appendix A.12) and a boxes-free setting using **SAM-2** proposals with Any Accuracy (Appendix A.13). Additionally, we clarify the **hyperparameter protocol** for Sec. 4.3 in the main text and Appendix A.3, and we make the **CSL p-value test definition** and the **FG-ratio baseline statistics** explicit in Appendix A.6 to prevent misinterpretation.
>
> **Clarifying the p-value in Sec. 4.1 (memorization reduction).** We define a sample as “easier to learn” if its cumulative sample loss is lower under FocL than under Standard for the same image (paired by sample index on the same training split), i.e., `CSL_FocL < CSL_Std`. In the top-1% Feldman–Zhang memorization cohort overlapping our Partition-A training set (`n = 820`), `819/820` samples improve (99.88%). We assess this with a **one-sided exact binomial (sign) test** under the null that improvement is equally likely as degradation (`H0: p = 0.5` for `CSL_FocL < CSL_Std`), which yields `p = 1.17 × 10^-244`. We additionally quantify the magnitude of improvement using a **paired t-test** on CSL differences (mean CSL drops from 204.37 to 72.22; `p = 1.09 × 10^-276`). We have added these test definitions (null/alternative and assumptions) to the Appendix to make the statistical evidence explicit.
>
> **Clarification on FG ratio and “contextual complexity” (Sec. 4.1).** In Sec. 4.1, **FG** is defined as the _foreground area fraction_ (ground-truth bounding-box area divided by full image area). A possible confusion is interpreting the memorized cohort mean `FG = 0.457` in isolation. Using our computed dataset-level statistics, the baseline mean FG over the full training splits is `0.470` (85K set) and `0.468` (482K set). Accordingly, the memorized cohort’s FG is **slightly lower than the dataset baseline**, i.e., these samples do **not** have unusually large foreground fractions. Instead, they follow the dataset’s **heavy-tailed FG distribution** (e.g., for the 85K split, `p10 ≈ 0.086`) and are, on average, **more background-dominated** than typical samples. This is **consistent with** our interpretation that many of these cases are challenging due to **contextual confounds** (e.g., cluttered backgrounds or co-occurring objects), rather than raw foreground pixel area alone.  This can be seen qualitatively in Figure 6 in the main manuscript and Figure 12 in Appendix. We thank the reviewer for highlighting this point and explicitly report these baseline statistics in the final manuscript to prevent misinterpretation.
>
> **Clarification on Hyperparameter Optimization (Sec. 4.3).** In Sec. 4.3, we compare the FocL single-crop variant against standard full-image training using an identical, fixed training protocol. Specifically, we use a standard ResNet-50 ImageNet training recipe (optimizer, batch size, augmentations, weight decay, and learning-rate schedule) and apply it unchanged to **both** Standard and FocL. We do **not** perform method-specific hyperparameter tuning or setting-specific schedule adjustments for FocL. Therefore, the observed differences in optimization stability (e.g., reduced gradient-norm magnitudes/spikes) and learning efficiency reflect the effect of object-centric input restructuring rather than hyperparameter tailoring. We have made this explicit in the main manuscript (Sec. 4.3) and Appendix to avoid ambiguity.

---

> ### Author Response · Authors · 2026-01-05
> **Rebuttal to Reviewer vWJm**
>
> **Clarifying the setup: FocL does not use segmentation as an input.** We believe the reviewer is using “segmentation” informally to mean _localization supervision_ (e.g., GT boxes or masks used to define object-centric inputs). A possible confusion is that FocL maps _(image, segmentation) → class_. That is not our setup. The classifier always consumes a standard RGB image tensor. Both Standard and FocL are trained with the same cross-entropy loss. Neither model receives segmentation masks, box coordinates, or any localization embedding as an additional input. Ground-truth boxes (and masks when used) are used only to define object-centric inputs (crop or masked image). They decide which RGB input is shown to the classifier. This is analogous to proposals produced by an external dorsal stream in an active-vision pipeline. At inference, we evaluate two settings. The first uses oracle ground-truth boxes. This corresponds to an oracle dorsal proposer. The second uses SAM proposals. SAM serves as a strong boxes-free dorsal module. These evaluations keep the classifier unchanged and isolate the effect of object-centric training.
>
> **Addressing the main weakness: intermediate baselines using localization supervision.** We agree this is an important point. We use architecture-preserving ResNet baselines (HardMask/MeanPad and RoIAlign) instead of coordinate embeddings that would change the classifier input space, and these baselines use the same gt boxes information without adopting FocL’s object-centric, jittered glimpse training to isolate context removal alone. First, we add **HardMask/MeanPad**, which neutralizes pixels outside the ground-truth box while keeping the object at its original full-frame scale. This isolates background removal without introducing object-centric, jittered glimpses. Second, we add a **RoIAlign** feature-pooling baseline that uses ground-truth boxes for feature pooling without image-space crop-and-resize. We report these baselines under **oracle GT-box** inputs (Appendix A.12) and also include an active vision inference pipeline using **SAM-2** proposals with Any Accuracy (Appendix A.13). We hope these additions address the reviewer’s concern and clarify that FocL’s gains are not explained only by providing localization supervision or by removing background context. We further point to the existing Table 1 diagnostic (Standard full vs Standard on oracle boxes) with qualitative support in Appendix Figs. 6–7.
>
> ## Oracle GT-box evaluation (Appendix A.12, Table 11)
>
> **Results.** Under oracle GT-box inputs, both intermediate baselines improve over Standard, but they remain well below FocL. RoIAlign and HardMask increase Top-1 by **+2.65 pp** and **+2.05 pp** over Standard, respectively. FocL (SC) reaches **71.05%** Top-1 and FocL (MC) reaches **72.23%**, which is **+10.15 pp** and **+11.33 pp** over Standard.
>
> **Explanation.** These modest gains are consistent with both baselines using localization to emphasize the foreground and reduce background reliance. However, **RoIAlign** pools features from a backbone feature map computed on the full image, so the pooled region can still carry contextual information due to large receptive fields and global feature computation. **HardMask** removes background pixels in image space, but the object remains at its original scale within the resized frame, so it does not introduce the same object-centric, jittered glimpse distribution that FocL trains on. FocL differs because it trains directly on **object-centric, jittered crops**, which shifts the input domain toward high-acuity object evidence and accounts for the remaining gap beyond background neutralization alone.
>
>
> ## SAM-2 proposals with Any Accuracy (Appendix A.13, Table 12)
>
> **Results.** Under SAM-2 proposal conditions, FocL remains the strongest approach across in-domain and shifted testing. RoIAlign can be competitive on ImageNet-V1 under its **native RoI pooling** mode (e.g., **73.05%**), but its performance degrades more sharply on ImageNet-V2, and the gap to FocL widens. When RoIAlign is evaluated using **localized object crops**, performance drops substantially (e.g., **59.45%** on V1 and **49.25%** on V2), indicating sensitivity to the inference mode and to distribution shift.
>
> **Explanation.**  FocL learns object-centric representations from **high-acuity foveal glimpses**, and its training includes **jittered foveal views** (randomized fixation/region sampling), which exposes the classifier to small but realistic variations in framing and scale. In contrast, the primary technical reason for the feature-pooling gap is that **RoI pooling operates on full-image feature maps with large receptive fields**. As a result, the pooled region features can still encode **dataset-specific contextual regularities** that assist performance on ImageNet-V1 but transfer less reliably to ImageNet-V2, leading to the more pronounced **V1→V2 accuracy degradation** observed in feature-pooling baselines.

---

> > ### Author Response · Authors · 2026-01-05
> > **Rebuttal to Reviewer vWJm**
> >
> > **(Nice-to-have sanity check) Weaker / locally-trained proposals instead of SAM:** We agree this would be a useful additional sanity check: replacing SAM with a weaker or locally trained proposal model could further reduce any concern about broad pretraining overlap. In our current setup, however, SAM is used **identically** for both Standard and FocL, so any proposal-model bias is shared and the observed gap reflects the effect of **object-centric training** rather than an advantage granted only to FocL; moreover, we also include **oracle/GT-box** diagnostics that remove dependence on any proposal model. Regarding exposure to ImageNet-V2/COCO, while some visual overlap is possible for web-scale models, SAM’s training protocol is class-agnostic and centered on SA-1B (11M images, 1.1B masks), which is distinct from the labeled COCO/ImageNet distributions.

---

> > > ### Comment · Reviewer_vWJm · 2026-01-06
> > > **Reply to Authors**
> > >
> > > > Clarifying the p-value in Sec. 4.1 (memorization reduction).
> > >
> > > Thanks, this makes sense.
> > >
> > > > Clarification on FG ratio and “contextual complexity” (Sec. 4.1).
> > >
> > > First, I apologize for any confusion as my original review mis-summarizes the authors' reasoning for memorization as the FG ratio being "relatively large" --- that is a mistake on my end and I mean to say "...relatively small". With this in mind, I don't understand the authors' rebuttal, so please let me know if I seem to be mis-interpreting anything further. The authors claim that the average FG ratio of the "memorization set" is only slightly less than the average FG ratio of the whole dataset but actually the dataset's FG ratio distribution has a heavy tail near 0 (and thus so does the memorization set's FG ratio?). I don't see why this would confirm your point, since:
> > >
> > > - If the memorization set has relatively few heavy-tail samples compared to the dataset, then actually the memorization set is actually inversely related with low FG ratio, which is not what you are claiming.
> > >
> > > - If the memorization set has many heavy-tailed samples compared to the dataset, then in order to balance out the FG ratios of the two sets to be nearly equal, many memorized samples should have relatively large FG ratio, which again contradicts the claim.
> > >
> > > - If they have similar proportions of heavy-tail samples then there seems to not be proof that FG ratio impacts memorization.
> > >
> > > Again I am not sure I understand the rebuttal and the additional writing in the paper so it would be great to clarify. (Also, the end of Appendix A6 still has rebuttal boilerplate: "We will explicitly report these baseline statistics in the final manuscript to prevent misinterpretation."
> > >
> > > > Clarification on Hyperparameter Optimization (Sec. 4.3).
> > >
> > > Thanks for clarifying that the hyperparameter settings are the same. It may be good to actually do some light hyperparameter optimization on both methods to see sensitivity to hyperparameters, but it is by no means a requirement.
> > >
> > > > Clarifying the setup: FocL does not use segmentation as an input.
> > >
> > > I think this is a matter of definitions. The backbone neural network is the same. But to use the FocL network, even for inference, it seems to me that one needs localization supervision, different from the usual image classifier. So while the literal neural network does not take a bounding box as input, one is still needed for each use of the model. So at least for each use of the model the signature is (image, bounding box) -> class. Similarly the training step does have the signature (image, class, bounding box) -> loss.
> > >
> > > > Addressing the main weakness: intermediate baselines using localization supervision.
> > >
> > > Thanks for including these comparisons, this is quite nice/interesting and I think a valuable addition to the paper.
> > >
> > > > (Nice-to-have sanity check) Weaker / locally-trained proposals instead of SAM
> > >
> > > Makes sense, I think we are on the same page here.
> > >
> > > **Overall**
> > >
> > > With the exception of the remaining concerns about the foreground-full image ratio connection to memorization, the rebuttal has strengthened all empirical claims made in the paper such that each is supported with evidence. Hence I have no other major concerns about accepting the work.

---

> ### Author Response · Authors · 2026-01-07
> **Clarification on FG and memorization**
>
> We apologize for the confusion caused by our earlier wording around FG. In the updated manuscript, **we do not use FG to explain memorization or as a proxy for contextual complexity**. FG is reported **only descriptively** in the qualitative visualizations and is **not** presented as a contribution or claim in the abstract or stated contributions. For completeness, Appendix A.6 reports FG statistics: on the 85K split, the mean FG is 0.470, while the FZ-intersected memorization cohort has mean FG 0.457. Concretely, **56.04%** of the 85K split has FG < 0.5, compared to **57.37%** of the FZ-intersected cohort (820 samples), a difference of **1.33 percentage points**, indicating FG does not meaningfully distinguish the two groups. Our memorization conclusion follows the core result in Sec. 4.1: we use Feldman–Zhang memorization scores to define a verifiably memorized cohort and show that FocL reduces CSL on this cohort with statistical significance (Appendix A.6 reports the hypotheses and exact p-value).
>
> We hope this clarification addresses the reviewer’s remaining concern. We thank the reviewer for the detailed comments, which helped us refine the manuscript and present our contributions more clearly.

---

### Review · Reviewer_bs49 · 2025-12-03

**Summary Of Contributions:**

This paper, inspired by the human visual system, investigates training paradigms for object recognition from foveated crops. The effectiveness of this approach is validated from multiple perspectives, including: 1) reduced model memorization requirements; and 2) enhanced robustness.

**Audience:**

Yes

**Audience Explanation:**

Yes, this paper explores a key issue in the field of object recognition.

Its research hypothesis—that objects need to learn mainly from the foreground to be robust and ignore the influence of spurious background information—is intriguing.

And the validation experiments are sufficiently thorough.

**Broader Impact Concerns:**

This work will not give rise to ethical implications.

**Claims And Evidence:**

Yes

**Claims Explanation:**

This paper provides highly comprehensive and multi-faceted experimental validation.
1. Fig. 1 provides an intuitive visualization of the impact of holistic learning.
2. Fig. 5, 6 effectively validate that the FocL training paradigm helps reduce model memorization.
3. Fig. 7 effectively demonstrates the contribution of the FocL training paradigm to model robustness.
4. Table 1 and 2 clearly demonstrate the effectiveness of this training method.

**Requested Changes:**

While the validation is extensive, a comparison with standard two-stage object detectors is necessary to validate your motivation, and proof that your "foreground-only" (hard crop) strategy is superior to the "localize-then-classify" (feature pooling) approach used in detection.

1. Dataset: ImageNet subset used in your paper, or the COCO dataset,
2. Baseline Method: Faster R-CNN or other methods your like, only if it is fair.
3. Evaluation: Compare your trained classifier with the classifier trained in these two-stage object detection methods (To ensure fairness, please evaluate the two-stage object detector's classification ability using the same ground-truth bounding boxes)

---

### Review · Reviewer_BEdk · 2025-12-21

**Summary Of Contributions:**

The paper introduces a supervised training strategy called FocL that is inspired by the human visual system's separation of dorsal (localization) and ventral (identification) pathways. The authors identify that standard deep learning models often rely on "spurious correlations" between labels and background clutter leading to memorizing hard examples rather than learning robust features. To mitigate this, FocL uses crops from foreground bounding boxes instead of full-image as training inputs, which they call foveated glimpses. The method simulates biological saccades by "jittering" the fixation point to generate multiple, slightly shifted views of the object, which are processed in the same mini-batch to enforce intra-class consistency.

Strengths:

* The paper provides compelling evidence for spurious correlations by showing that standard models perform worse when given ground-truth oracle crops, proving their reliance on background context.

* The method works with SAM which opens up opportunities for integrating it with foundation models in general for object proposals.

* The paper includes empirical results from extensive experiments in several dimensions such as learning dynamics, robustness, and generalization.

Weaknesses:

* The paper claims "learning with less data" but it doesn't account for higher cost of bounding boxes compared to simple labels for images.

* The authors ran baselines on full images and their method on crops. It's not a fair comparison. At least the authors didn't point out how much advantage they already have by starting from crops..

* Again, empirical results are between full images and crops. We're missing intermediate baselines like saliency-based cropping or copy-paste augmentation.

**Audience:**

Yes

**Audience Explanation:**

The paper addresses core challenges of deep neural networks, such as robustness, memorization, and data efficiency. These are central and interesting to the TMLR community. The proposed method combines specialist classifier with a foundation model, SAM, is also relevant to current trends.

**Broader Impact Concerns:**

No concern.

**Claims And Evidence:**

Yes

**Claims Explanation:**

The authors's main areas of claims are well supported.

* Memorization: They use cumulative sample loss and the Feldman & Zhang memorization score to show that hard samples become easier to learn when background clutter is removed.

* Generalization: The paper shows that the proposed method achieved significant gains in top-1 accuracy on ImageNet-V2 and COCO.

* The paper also claims smoother optimization and faster convergence which are supported by results from tests with ImageNet, in 4.3.

* The paper also claims that their FocL learns "more stable features". It's supported by evidence such as tighter t-SNE clusters in Figure 2 and larger $l2$ adversarial distances in 4.1.

**Requested Changes:**

About the efficiency claims. Authors claim that the proposed method uses "56% less training data" and thus offers "efficient visual recognition". This is misleading because it ignores the high cost of bounding box annotations. The authors should acknowledge that although they can be more efficient by using fewer samples, the annotation efficiency is actually going lower compared to standard class-label training.

About related work: The literature review in the paper contrasts FocL with complex RL policies or unsupervised discovery. The paper should mention saliency-based data augmentation which is a simpler and also automated method for breaking background dependence.

About adversarial evaluation in 4.1. The authors attack the standard model on full images and FocL on crops. Since cropping out the background pixels making it harder to hide perturbations. The paper should acknowledge that cropping itself is a contributing factor to the increased perturbation distance, in addition to the improved feature stability, unless the authors don't agree with me.

About batch normalization. In section 3, the paper goes that glimpses from the same image are in the same mini-batch, which doesn't fit a typical assumption that samples in the same mini-batch are i.i.d. We need an explanation on whether this correlation affected batch normalization statistics, either positively or negatively.

---

> ### Author Response · Authors · 2026-01-04
> **Rebuttal for Reviewer BEdk**
>
> Thank you for the thoughtful feedback. We have revised the manuscript to address your concerns regarding **(i)** data-efficiency claims under bounding-box supervision, **(ii)** fairness and interpretation of adversarial robustness comparisons across full images vs. crops, **(iii)** missing intermediate baselines, and **(iv)** correlated multi-glimpse batches and BatchNorm statistics. All changes are reflected in the updated manuscript (**marked in purple**), with additional clarifications and ablations referenced in the Appendix.
>
> ### 1) Data efficiency claim and supervision/annotation cost
>
> **Concern:** The “56% less training data” statement may be misleading since FocL assumes object localization supervision (e.g., bounding boxes), whose annotation cost is not directly comparable to image-level labels.
>
> **Response / Changes:** We agree and revised the wording to avoid implying reduced annotation cost. Specifically, we updated the **Abstract** and **Contributions** to state **“56% fewer training images given access to object localization (bounding boxes or proposals).”** We also added a concise clarification in **Sec. 4.3** that this efficiency is measured **under the localization-supervised setting** and does not claim reduced human labeling cost (in our experiments, boxes are human-annotated). Finally, we note that localization can also be provided by **automated proposal models** (e.g., SAM). We additionally clarify the intended interpretation: FocL’s efficiency reflects **extracting more value per localized image** by enforcing object–label alignment and reducing reliance on background-correlated cues, which is consistent with reaching higher accuracy using substantially fewer training images in this setting.
>
> ### 2) Saliency-based data augmentation to reduce background dependence
>
> **Concern:** The paper should mention saliency-based data augmentation as a simpler automated mechanism for reducing background dependence.
>
> **Response / Changes:** We added a detailed discussion of saliency-guided augmentation/cropping in the **Appendix (Expanded Related Work)** and clarified how FocL differs from these approaches. In particular, FocL uses **bounding-box–driven foveated jitter** to ensure strong **object–label alignment**, which is important for our downstream analyses of memorization and stability. While saliency/activation maps could in principle be used to propose foveated crops, these proposals are **less controlled for object–label alignment** and can highlight background-correlated regions or yield crops with substantial background overlap, which may weaken the object-aligned supervision that motivates our formulation.
>
> ### 3) Intermediate baselines
>
> **Concern:** The evaluation lacks intermediate baselines (e.g., saliency cropping or copy-paste style augmentation) between full-image training and foveated cropping.
>
> **Response / Changes:** We discuss saliency-based approaches in related work **and** add two intermediate object-centric controls that are directly comparable under the same GT boxes. Specifically, Appendix **A.10–A.13** introduces baselines that isolate the role of input restructuring: **HardMask/MeanPad** (pixel-space context removal without foveation) and a **RoIAlign-style** localization alternative (feature pooling given GT boxes).  These controls also directly address related fairness requests raised by other reviewers (HardMask for vWJm; RoIAlign/feature pooling for bs49). We also reference these appendix sections in the main manuscript and direct the reader there for the full setup, results, and conclusions.

---

> > ### Author Response · Authors · 2026-01-04
> > **Rebuttal for Reviewer BEdk**
> >
> > ### 4) Adversarial robustness: full images vs. crops and “hiding perturbations in background”
> >
> > **Concern:** Attacking the standard model on full images and FocL on crops may increase the measured adversarial distance simply because background pixels are removed, making perturbations harder to “hide.”
> >
> >
> > **Response / Changes:** As already stated in **Sec. 4.1 (Adversarial Distance)**, we perform PGD on each method’s **native 224×224 input representation** to match its training regime and avoid conflating robustness with a **distribution shift** from attacking a non-native input representation. This directly reflects the “**full images vs. crops**” comparison: the Standard baseline is attacked on the full 224×224 image it consumes, while FocL is attacked on the **224×224 foveated (oracle) crop** it consumes at test time; no resizing is performed inside the PGD loop. Accordingly, the reported ℓ2 distance quantifies robustness of the **deployed end-to-end FocL pipeline** (object-centric input + classifier), rather than an input-matched, pixel-wise full-frame robustness comparison under identical content.
> >
> > We also acknowledge your point about “hiding” perturbations in background regions and provide explanation in **Key Takeaways (Sec. 4.1)**. Reduced dependence on spurious background is an intended **feature** of foveated perception: the goal is to ignore clutter and force decisions to rely on object-relevant evidence. This interpretation is consistent with prior active-vision robustness analyses [1] that **associate** increased adversarial stability **with** foveation and fixation-based processing rather than passive full-frame processing.
> >
> > ### 5) Correlated glimpses within a mini-batch and BatchNorm statistics
> >
> > **Concern:** Grouping multiple glimpses from the same image in the same mini-batch violates an i.i.d. assumption and may affect BatchNorm statistics positively or negatively.
> >
> > **Response / Changes:** We added a brief note in **Sec. 3.1 (Methodology)** clarifying that BatchNorm is applied in the standard way over the **concatenated crop batch** (across images and glimpses). While glimpses from the same image are correlated, each glimpse is generated with **independent foveated jitter** (and standard ImageNet augmentations), so the batch does not “collapse” to near-duplicates and retains substantial within-batch diversity. Empirically, **Appendix Tables 6–7** show the multi-crop variant (k=3) is consistently better than single-crop, suggesting correlated multi-glimpse batching does not harm BN in practice and can improve generalization.
> >
> > [1]. "On Inherent Adversarial Robustness of Active Vision Systems", TMLR 2025

---

> > > ### Comment · Reviewer_BEdk · 2026-01-05
> > > **No more concerns from me**
> > >
> > > I appreciate the thorough response. I think the paper is complete and I have no more concerns. I will be waiting for the final version of the manuscript (without editing marks like blue and red texts). Please also proofread the manuscript since the edit introduced typos like the one right before 4. Experiments: "selection criteria, is provided in Appendix A.2" doesn't have a period to end the sentence.

---

### Decision · Action_Editor_fQqD · 2026-01-22

**Recommendation:** Accept as is

**Additional Comments:**

Please double-check that all reviewers' comments and changes have been incorporated into the camera-ready version, which should follow TMLR's template carefully. The camera-ready version will be checked for compliance and to ensure that all comments have been incorporated.

**Audience:**

Yes

**Audience Explanation:**

The work targets core themes for TMLR, such as robustness, memorisation, and data efficiency, and connects object‑centric learning with a practical active‑vision pipeline. The breadth of experiments across ImageNet‑V1/V2 and COCO, plus analyses of learning dynamics and adversarial stability, should interest researchers in representation learning, recognition, and robustness.

**Claims And Evidence:**

Yes

**Claims Explanation:**

The paper introduces a straightforward, object‑centric training strategy (called FocL) and supports its claims with extensive empirical evidence covering memorisation, robustness, generalisation, and efficiency (under localisation supervision). The authors included intermediate baselines, i.e. RoIAlign (feature pooling) and HardMask/MeanPad (background neutralisation without foveation), to determine whether improvements are solely due to context removal. Using oracle GT‑box inputs (2K evaluation), RoIAlign (Top‑1 63.55%) and HardMask (62.95%) outperform Standard (60.90%), but still fall short of FocL single‑/multi‑crop (71.05% / 72.23%), demonstrating that the advantage is not just related to context suppression but involves object‑centred, jittered foveation during training.

On proposal‑based inference with SAM‑2, FocL attains the strongest Any Accuracy on ImageNet‑V1 (75.05% / 75.82%) and ImageNet‑V2 (64.80% / 65.80%) versus Standard localised-crops (68.08% / 57.50%) and RoIAlign (best 73.05% / 58.75% under native RoI pooling), supporting robustness under natural distribution.

Concerns raised by reviewers (fairness of adversarial comparisons, BatchNorm with correlated glimpses, data‑efficiency wording) are substantively addressed in the revised text and appendix.